



# Altitudinal Control of Isotopic Composition and Application in Understanding Hydrologic Processes in the mid Merced River Catchment, Sierra Nevada, California, USA

Fengjing Liu[1, a], Martha H. Conklin[2], and Glenn D. Shaw [3]

[1] College of Forest Resources and Environmental Science,
Michigan Technological University, Houghton, MI
[2] Sierra Nevada Research Institute & School of Engineering,
University of California, Merced, CA
[3]Department of Geological Engineering, Montana Tech of the University of Montana, Butte, MT

**ª Corresponding Address†**

Fengjing Liu

College of Forest Resources and Environmental Science

Michigan Technological University

1400 Townsend Drive

Houghton, MI 49931

Phone: 906-487-1089

Email: fliu7@mtu.edu





**Abstract.** Mountain snowpack has been declining and more precipitation has fallen as rainfall than snowfall, particularly in the US West. Isotopic composition in stream water, springs, groundwater, and precipitation was examined to understand the impact of declining snowpack on hydrologic processes in the mid Merced River catchment (1,873 km$^2$), Sierra Nevada, California, USA. Mean isotopic values in small tributaries (catchment area < 122 km$^2$), rock glacier outflows and

groundwater from 2005-2008 were strongly correlated with mean catchment elevation ($R^2 = 0.96$ for $\delta^2$H, $n = 16$, $p < 0.001$), with an average isotopic lapse rate of -1.9‰/100 m for $\delta^2$H and -0.22‰/100 m for $\delta^{18}$O in meteoric water. The lapse rate did not change much over seasons and was not strongly affected by isotopic fractionation. A catchment-characteristic isotopic value was thus established for each sub-catchment based on the relation between isotopic composition and

the mean catchment elevation to elucidate hydrometeorologic and hydrologic processes. Compared to Tenaya Creek without water falls, flow and flow duration of Yosemite Creek are much more sensitive to temperature increase due to a strong evaporation effect caused by waterfalls, suggesting possible prolonged dry-up period of Yosemite Falls in the future. Groundwater in the Yosemite Valley (~900-1,200 m) was recharged primarily from the upper

snow-rain transition zone (2,000-2,500 m), suggesting its strong vulnerability to shift in snow-rain ratio. The information gained from this study helps advance our understanding of hydrologic responses to climate change in snowmelt-fed river systems.

**Key Words:** Stable isotopes, isotopic lapse rate, groundwater recharge, snow-rain transition,
climate change, Yosemite Falls, Merced River







## 1. Introduction

With an increase in global temperature, snow cover extent has decreased in the Northern Hemisphere, especially in spring (*Vaughan et al*., 2013). In the mountain regions of the U.S. West, less precipitation falls as snow (e.g., *Mote et al*., 2005; *Knowles et al*., 2006) and the melting of snow starts earlier (e.g., *Stewart et al.*, 2004). Even without any changes in precipitation amount, observations and modeling results have shown that less snow and earlier snowmelt lead to a shift in peak river runoff toward late winter and early spring, away from summer when water demand is highest (e.g., *Dettinger and Cayan*, 1995; *Barnett et al*., 2005; *Stewart et al*., 2005). A decrease in snow to rain ratio also reduces groundwater recharge within the mountain block (*Earman* et al., 2006; 2011; *Penna et al*., 2014). It is anticipated that these changes in snow condition and subsequent responses of stream flow and groundwater recharge are strongest in the snow-rain transition zone (e.g., *Tennant et al*., 2015), which is 1,500-2,500 m in Sierra Nevada, California based on *Hunsaker et al*. (2012).

However, our present knowledge of watershed hydrology is still not sufficient to fully understand the impact of these changes on stream flow and groundwater recharge (*Kundzewicz et al*., 2007; *Alley*, 2001; *Fayad et al*., 2017). Particularly for catchments with a Mediterranean climate such as those in Sierra Nevada, California and Europe, where precipitation is little after the snowmelt season in spring and early summer, it is unclear how the changes in snow condition in spring affects baseflow (stream flow after snowmelt period or low flow) in late summer and fall (*Fayad et al*., 2017). This problem is primarily caused by lack of accurate hydrologic measurements in mountains (*Bales et al*., 2006) and adequate techniques to determine groundwater recharge generated from snowmelt and rainwater (*Wilson and Guan*, 2004; *Manning & Solomon*, 2005; *Manning & Caine*, 2007).

The stable isotopes of oxygen and hydrogen in the water molecule have become an important tool for studies on atmospheric circulation, palaeoclimate and watershed hydrology (*e.g*., *Araguas-Araguas et al*., 2000). In meteorology, stable isotopes have been applied to understand the trajectory of air moisture (e.g., *Friedman et al.*, 2002; *Peng et al*., 2016), as the isotopic composition of precipitation is largely controlled by the water vapor source, the formation temperature of precipitation, and the relative fraction of water vapor removed from the atmosphere (*Gat*, 1996). Stable isotopes have also been used to infer paleoclimate and paleohydrology using proxy records such as ice cores (e.g., *Thompson et al*., 2000) and to reconstruct paleoelevations



based on the elevation gradient (lapse rate) of stable isotopic composition in modern precipitation (e.g., *Poage and Chamberlain*, 2001). In watershed hydrology, the isotopic composition has been widely applied to study the origin and dynamics of stream water and groundwater across varying

climates and land covers from snow-dominated catchments in high elevations to forested catchments in temperate regions (*e.g*., *Kendall and McDonnell*, 1998; *Wen et al.*, 2016; *Penna et al.*, 2017). The distinctness of isotopic composition among source waters (endmembers) is the basis for the studies of watershed hydrology and allows identification and even quantification of the contributions of source waters to stream flow (e.g., *Sklash et al*., 1976; *Liu et al*., 2004; *Penna*

*et al.*, 2016).

       The success of the above applications hinges on our understanding of the processes or factors that control the isotopic composition in the studied subject (e.g., stream water, groundwater, water vapor, and snow). Any of the intrinsic relationship between isotopic composition and environmental variables (e.g., air temperature and elevation) may be best suited

for one application, but may create a barrier for another application. In paleoelevation studies, for example, an accurate isotope-elevation relationship in meteoric water can be used to place numerical constraints on the topographic development of ancient mountain belts or plateaus with the information on past changes in the isotopic composition of precipitation preserved in pedogenic or authigenic minerals (e.g., *Poage and Chamberlain*, 2001; *Mulch et al.*, 2006; *Hren et al*., 2009).

However, the variability of isotopic composition with elevation may complicate the application of a mixing model to determine endmember contributions to stream flow, particularly for large catchments where the variation of isotopic composition due to elevation may mask the variation from one endmember to another. In tracing tropical Andean glacier melt, for example, *Mark and McKenzie* (2007) had to eliminate the effect of elevation on isotopic composition of stream water

before applying a mixing model to estimate the contributions of glacier melt to stream flow at various catchment scales. It is thus imperative to fully comprehend the controls on stable isotopes in waters before they are applied in any disciplines.

       As the first step in an ongoing effort to quantify how change in the snow-rain proportion affects stream flow and groundwater recharge in a snowmelt-fed river system, the objectives of

this study were to understand the processes or factors that control the spatiotemporal variability of isotopic composition in precipitation, stream water and groundwater and how such information could be used to advance our understanding of hydrometeorologic and hydrologic processes in a





snowmelt-fed river system. This study was conducted in the Merced River above Briceburg (mid
Merced River catchment) (Figure 1), a representative snowmelt-fed river system for the central

and southern Sierra Nevada, California. Isotopic data were acquired from precipitation, springs,
groundwater, and stream water during the 2005-2008 period, which includes a very wet year
(2006) and a very dry year (2007). The data from such a period thus provides us with an excellent
opportunity to examine the variability of stable isotopic composition in surface water and
groundwater with precipitation extremes in the mid Merced River catchment.


## 2. Research site

The study was conducted in the mid Merced River catchment above Briceburg, including
Yosemite Valley (Figure 1). The mid Merced River catchment drains 1,873 km$^2$, and ranges in
elevation from 346 m at Briceburg to 3,993 m eastward at the crest. The drainage is relatively

undisturbed by human activities such as dams, much of it within Yosemite National Park (YNP).
The mid Merced River was designated a Wild and Scenic River in 1987 by the U.S. Congress.

The mid Merced River catchment is characterized by a Mediterranean climate, with
moderately wet, cold winters and dry, warmer summers. The mean annual precipitation at the
Yosemite Valley (Figure 1) has been 916 mm, based on data from 1917 to 2008. Precipitation in

the region occurs primarily from October to April, mainly as snow above 2,500 m and rain below
1,500 m, as shown by meteorological data at a neighboring site in the southern Sierra Nevada,
about 100 miles south to the Merced River (*Hunsaker et al.*, 2012). Precipitation from May to
October accounted for only 25% of the annual mean precipitation.

Like most of the Sierra Nevada range, the mid Merced River catchment is underlain by

granitic rocks of the Sierra Nevada batholith. Most of the rocks are part of the Tuolumne Intrusive
Suite, a group of four concentrically arranged plutonic bodies, within which are all granites and
granodiorites (*Bateman*, 1992). Vegetation covers approximately 45% of the catchment and
includes a red fir forest that grades into a mixed subalpine forest above 2,750 m (*Rundel et al.*,
1977). Above the timberline (~3,200 m), the vegetation consists of low-lying tundra plants and

alpine meadow vegetation. Surficial deposits cover about 20% of the catchment above Happy Isles
and valleys are covered primarily by glacial tills that occur in valley bottoms as lateral and
recessional moraines (*Clow et al.*, 1996). Wells drilled in the Yosemite Valley indicate that the
deposit is about 300 m, consistent with *Gutenburg et al.* (1956), which is dominated by



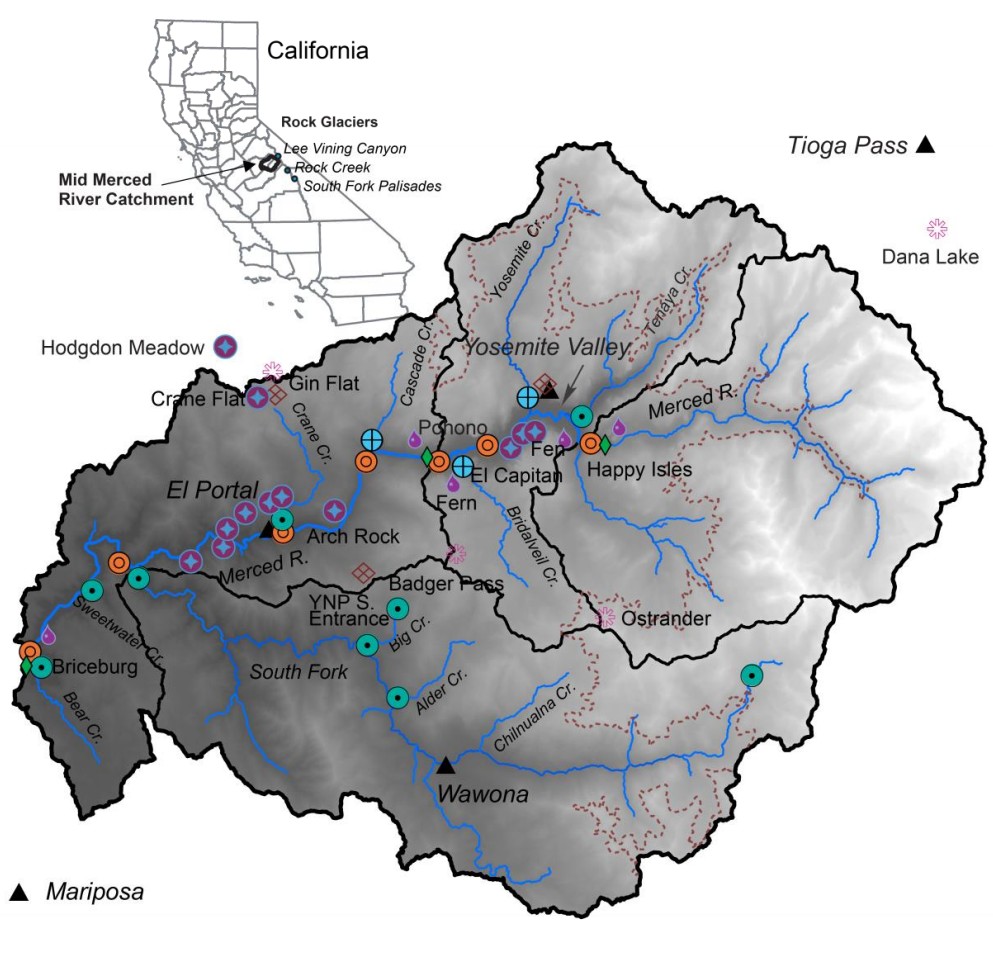

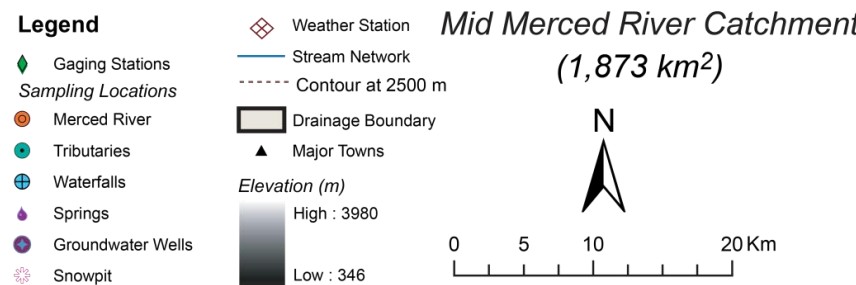

Figure 1. Sampling locations for snow, stream water, spring water and groundwater in the mid Merced River catchment, along with stream gages and meteorological stations, topography, stream network and drainage boundary. The inset map shows the locations of the mid Merced River catchment in California and rock glaciers outside the catchment. The elevation contour at 2,500 m (brown dashed line) is also marked to show the upper boundary of the snow-rain transition zone.


unconsolidated sands from land surface to about 20 m below, mainly silt from 20 m to 70 m, granitic gravels in silt from 70 m to 80 m, and chiefly boulders and sands below 80 m. The deposit in the lower section of the catchment from El Portal to Briceburg is approximately 20 m in depth, consisting of gravels, cobbles, decomposed granite, sand and silt.

**3. Methods**

**3.1. Hydrologic and meteorological data**

        Hydrologic and meteorological data were downloaded from the California Data Exchange Center (CDEC; https://cdec.water.ca.gov; access verified on August 17, 2023). Stream flow was measured at Happy Isles and Pohono Bridge (data also available for both sites at

https://waterdata.usgs.gov/usa/nwis; access verified on August 17, 2023) (Figure 1) by the United States Geological Survey (USGS) and daily mean discharges were used in the study. Happy Isles is a USGS Hydrologic Benchmark Network site; this network was developed, in part, for its utility as a long-term monitoring network designed for detection of trends in stream flow and chemistry in response to changes in climate (*Mast and Clow*, 2000). Note that stream flow at Briceburg was

measured by the Merced Irrigation District. The stage sensor at Briceburg is located inside of a stilling well from which water is pumped out to supply water for the city of Mariposa, which may cause water level to drop several feet during short periods. The stream flow data at Briceburg was thus used with care in this study. Precipitation was measured at Yosemite Valley, Gin Flat and Wawona by the Yosemite National Park and the California Department of Water Resources. Snow

depth was measured by snow courses, operated by the California Department of Water Resources and U.S. Natural Resource Conservation Service, and the daily values at Gin Flat, Ostrander and Tioga Pass were selected. Daily snow water equivalent (SWE) data was not available for all stations and thus snow depth was used in this study. Snow depth data from other stations in the catchment was not selected because daily values were not available. Tioga Pass, located just

outside the catchment, was selected because it is the only one located above 3,000 m in the region.

**3.2. Sample collection**

        Samples of stream water, groundwater, and spring water were collected from the 2005-2008 period through extensive field campaigns in the mid Merced River catchment (Figure 1 and

Table 1). Stream water samples were collected weekly to biweekly at about twenty locations along





the Merced River, including gages at Happy Isles, Pohono Bridge and Briceburg, and major tributaries. Note that samples of Merced River at Cascade Picnic Area were collected from a spot right after the confluence of Cascade Creek (Figure 1). The Merced River channel is wide-open in that section and the sampling spot is on the same side as Cascade Creek. Water from Merced River


Table 1. Mean $\delta^{18}O$ and $\delta^2H$ values with $\pm1\sigma$ in streams, glacier outflows, spring water, groundwater, and precipitation in the mid Merced River catchment and vicinity, along with catchment characteristics.

| | | Sample | | | | Catchment | Catchment Elevation | | $\delta^{18}O$ Values | | $\delta^2H$ Values | |
| Type | Locations | Start Date | End Date | Number $n$ | Elevation (m) | Area (km$^2$) | Mean (m) | Max (m) | Mean (‰) | $\pm1\sigma$ (‰) | Mean (‰) | $\pm1\sigma$ (‰) |
|---|---|---|---|---|---|---|---|---|---|---|---|---|
| | Happy Isles | 11/11/2005 | 8/7/2008 | 68 | 1251 | 468 | 2743 | 3993 | -13.8 | 0.9 | -102.4 | 5.1 |
| | El Capitan | 11/11/2005 | 8/7/2005 | 49 | 1206 | 744 | 2624 | 3993 | -13.4 | 0.7 | -98.8 | 4.1 |
| | Pohono Bridge | 5/19/2006 | 8/7/2008 | 64 | 1179 | 833 | 2580 | 3993 | -13.3 | 0.7 | -98.0 | 4.3 |
| Merced River | Cascade Picnic Area | 11/11/2005 | 7/22/2008 | 37 | 1040 | 902 | 2539 | 3993 | -12.7 | 0.7 | -91.8 | 4.4 |
| | El Portal | 9/1/2006 | 7/22/2008 | 35 | 605 | 961 | 2483 | 3993 | -13.1 | 0.7 | -96.5 | 4.2 |
| | South Fork Confluence | 3/30/2006 | 7/22/2008 | 33 | 424 | 1087 | 2350 | 3993 | -12.9 | 1.0 | -93.1 | 5.3 |
| | Briceburg | 11/11/2005 | 7/22/2008 | 54 | 346 | 1873 | 2067 | 3993 | -12.4 | 1.1 | -90.5 | 7.5 |
| | Tenaya Creek | 11/6/2006 | 8/7/2008 | 43 | 1212 | 122 | 2528 | 3310 | -13.1 | 0.6 | -95.9 | 2.6 |
| | Yosemite Creek | 11/11/2005 | 8/7/2008 | 50 | 1249 | 109 | 2516 | 3294 | -12.0 | 1.8 | -89.2 | 8.8 |
| | Bridalveil Creek | 11/11/2005 | 8/7/2008 | 48 | 1284 | 64 | 2232 | 2837 | -12.1 | 0.7 | -87.2 | 3.6 |
| | Cascade Creek | 11/11/2005 | 6/6/2008 | 38 | 1143 | 50 | 2228 | 2736 | -12.0 | 0.6 | -85.0 | 3.6 |
| | Crane Creek | 11/11/2005 | 7/22/2008 | 37 | 602 | 46 | 1621 | 2163 | -11.4 | 0.6 | -79.6 | 2.7 |
| Tributaries | South Fork | 11/11/2005 | 8/4/2008 | 40 | 425 | 623 | 1857 | 3575 | -11.9 | 1.2 | -85.6 | 7.0 |
| | Sweetwater Creek | 8/21/2006 | 7/22/2008 | 32 | 375 | 18 | 1058 | 1408 | -10.2 | 0.4 | -70.4 | 1.6 |
| | Bear Creek | 9/1/2006 | 6/13/2008 | 29 | 348 | 58 | 913 | 1409 | -9.0 | 0.5 | -64.2 | 2.1 |
| | Alder Creek | 7/16/2008 | 8/5/2008 | 6 | 1099 | 39 | 1806 | 2446 | -12.0 | 0.3 | -85.0 | 0.8 |
| | Big Creek at Fish Camp | 7/16/2008 | 8/4/2008 | 6 | 1515 | 44 | 1946 | 2649 | -12.3 | 0.4 | -86.1 | 0.9 |
| | Big Creek at South Fork | 7/16/2008 | 8/5/2008 | 6 | 1203 | 80 | 1798 | 2649 | -11.8 | 0.4 | -83.3 | 0.7 |
| | Headwater of South Fork | 8/4/2008 | 8/4/2008 | 1 | 2754 | 8 | 2969 | 3550 | -13.0 | N/A | -101.3 | N/A |
| | Lee Vining Canyon | 7/21/2006 | 7/21/2006 | 1 | 2965 | 1 | 3271 | 3531 | -15.3 | N/A | -115.5 | N/A |
| Rock Glaciers | South Fork of Palisade | 7/20/2006 | 10/7/2007 | 6 | 3289 | 2 | 3624 | 4067 | -15.8 | 0.6 | -117.1 | 5.1 |
| | Rock Creek | 8/18/2006 | 7/15/2007 | 4 | 3568 | 1 | 3772 | 4101 | -16.6 | 0.7 | -120.2 | 4.3 |
| | Happy Isles | 4/6/2006 | 8/7/2008 | 39 | 1210 | | | | -13.5 | 0.3 | -99.0 | 2.0 |
| Springs | Fen | 8/21/2006 | 8/7/2008 | 29 | 1109 | | | | -13.7 | 0.3 | -98.3 | 1.3 |
| | Fern | 11/11/2005 | 8/7/2008 | 55 | 1199 | | | | -12.3 | 0.4 | -86.8 | 1.3 |
| | Drinking Fountain | 4/6/2006 | 7/22/2008 | 25 | 372 | | | | -9.6 | 0.3 | -67.6 | 1.1 |
| | Valley Well 1 | 6/21/2005 | 7/15/2008 | 5 | 1188 | | | | -12.8 | 0.2 | -94.1 | 1.5 |
| | Valley Well 2 | 6/21/2005 | 7/15/2008 | 5 | 1180 | | | | -12.5 | 0.2 | -91.9 | 1.1 |
| | Valley Well 4 | 6/21/2005 | 7/15/2008 | 5 | 1183 | | | | -12.7 | 0.2 | -93.5 | 1.0 |
| | Arch Rock | 6/21/2005 | 10/24/2007 | 4 | 933 | | | | -12.4 | 0.1 | -89.5 | 1.2 |
| | Crane Flat | 6/21/2005 | 7/15/2008 | 5 | 1994 | 0.2 | 2011 | 2027 | -12.4 | 0.1 | -85.9 | 0.7 |
| Groundwater | Hodgdon Meadow | 6/21/2005 | 7/15/2008 | 5 | 1407 | 4 | 1542 | 1836 | -11.5 | 0.2 | -81.5 | 0.7 |
| | El Portal Well 2 | 6/21/2005 | 7/15/2008 | 5 | 565 | | | | -10.9 | 0.3 | -80.4 | 2.3 |
| | El Portal Well 3 | 6/21/2005 | 7/15/2008 | 5 | 571 | | | | -11.0 | 0.4 | -81.2 | 4.3 |
| | El Portal Well 4 | 6/21/2005 | 7/15/2008 | 5 | 561 | | | | -11.9 | 0.6 | -87.2 | 6.0 |
| | El Portal Well 5 | 6/21/2005 | 7/15/2008 | 5 | 544 | | | | -11.5 | 0.5 | -83.9 | 3.9 |
| | El Portal Well 6 | 6/21/2005 | 7/15/2008 | 5 | 567 | | | | -12.4 | 0.3 | -90.9 | 2.1 |
| | El Portal Well 7 | 6/21/2005 | 7/15/2008 | 5 | 563 | | | | -12.6 | 0.5 | -92.7 | 2.9 |
| | Gin Flat | 4/27/2006 | 4/27/2006 | 23 | 2150 | | | | -11.4 | 2.0 | -82.4 | 15.6 |
| Snowpits | Badger Pass | 3/27/2006 | 3/31/2006 | 13 | 2226 | | | | -13.2 | 1.2 | -93.9 | 10.5 |
| | Ostrander | 3/29/2006 | 3/29/2006 | 25 | 2500 | | | | -14.6 | 2.5 | -106.5 | 21.0 |
| | Dana Lake | 8/18/2005 | 8/18/2005 | 3 | 2926 | | | | -14.7 | 1.5 | -105.5 | 12.1 |
| Precipitation | NADP | 11/14/2006 | 4/24/2007 | 10 | 1393 | | | | -11.5 | 2.5 | -80.2 | 17.8 |


and Cascade Creek may not be well mixed at the sampling spot due to the short distance to the
confluence, but a well-mixed spot cannot be established due to local landscape, safety and logistic
issues. In addition, an earlier study showed that this area is a groundwater discharge zone (*Shaw et al.*, 2014). So, data from this site was used and interpreted cautiously.

Water samples were collected from four springs located near the Merced River between
Happy Isles and Briceburg (Figure 1), with a frequency varying from weekly to monthly. Water
samples were also collected bi-annually during snowmelt and off-snowmelt seasons from 2005 to
2008 from drinking water wells located in Yosemite Valley, El Portal, Crane Flat and Hodgdon
Meadow (Figure 1). The depths of wells range from 100 to 120 m in Yosemite Valley and from
20 to 30 m at El Portal. Information on the depth of other wells was not available. Samples were
taken directly from the sampling ports.

Water samples were also collected at the outflows of three rock glaciers at the South Fork
of Palisade River, Rock Creek and Lee Vining Canyon, just outside the mid Merced River
catchment (Figure 1). These samples were collected 1-4 times from July 2006 to October 2007.

Snow and rain samples were collected at the National Atmospheric Deposition Program
(NADP) site (Site ID = CA99, elevation = 1,393 m) in Yosemite National Park from November
2006 through April 2007. These samples were collected from a rain gage right after storms and
only from relatively large storms when there was enough water left over after the NADP samples
were collected. These samples were from snowfall, rainfall, and a mixture of snowfall and rainfall
based on the collector's notes.

Three snowpits were excavated near the maximum snow accumulation in late March and
early April 2006 at Badger Pass, Gin Flat and Ostrander near Yosemite Valley (Figure 1; Table
1). The depth of snowpits ranges from 1.5 to 2.5 m. Snow samples were collected continuously
every 10-cm throughout the entire pit at Badger Pass, Ostrander and Gin Flat. Three snow core
samples were collected in summer 2005 at Dana Lake, just below the crest on the eastern side of
Sierra Nevada and outside the mid Merced River catchment. Snow samples were stored in plastic
bags pre-rinsed with deionized water (DI) and washed by sampling snow at the time of collection.
Snow samples were melted at room temperature immediately upon arrival at the laboratory.

All liquid water samples were stored in 30-mL glass vials with snap-on caps. All samples
were checked for the absence of air bubbles. After collection, samples were transported to the
University of California, Merced and kept refrigerated at 4 °C until analysis.





**3.3. Sample analysis**

The stable isotope ratios ($^{18}O/^{16}O$ and $^2H/^1H$) of all samples are expressed as δ (per mil, expressed as ‰) variation in the ratio of the sample relative to Vienna Standard Mean Ocean Water (VSMOW). Samples collected in 2005 and 2006 were analyzed at the University of California, Berkeley, using a VG PRISM isotope ratio mass spectrometer, with a precision of 0.05‰ for $\delta^{18}O$

and 0.3‰ for $\delta^2H$. Samples collected in 2007 and 2008 were analyzed using a Los Gatos LTD100 Isotopic Analyzer at the University of California, Merced. This analyzer is based on continuous laser absorption spectroscopy (LAS). The precision of this instrument was comparable to conventional mass spectrometer (*Wang et al.*, 2009a), with our data showing 1σ precision better than 0.2‰ for $\delta^{18}O$ and 0.3‰ for $\delta^2H$, consistent with *Berman et al.* (2009). The precision was

slightly better for $\delta^2H$ than for $\delta^{18}O$ because the measurement of $^{18}O/^{16}O$ was more sensitive to varying room temperatures (Personal Communications with Los Gatos Company, 2009). For this reason, $\delta^2H$ values were primarily presented in this study where both $\delta^{18}O$ and $\delta^2H$ values did not have to be used.

**3.4. Drainage delineation**

Drainage above a gage or a sampling point was delineated using 30-m DEM following the standard procedure described in the ArcGIS 10.0 manual (ESRI Inc.). The 30-m DEM data were acquired from a USGS web site (http://seamless.usgs.gov; now https://www.usgs.gov/the-national-map-data-delivery/gis-data-download as of August 17, 2023). The geographic location of

a gage or a sampling point was used as a pour point. After the delineation, the mean elevation for the drainage was calculated as arithmetic average of all raster grid elevations within the drainage.

**4. Results**

**4.1. Hydrometeorology**

Hydrologic conditions were very different in water years (October 1 in the previous year to September 30) 2006, 2007 and 2008 (all referring to water years hereinafter; otherwise stated). Precipitation and snow depth were much higher in 2006 than in 2008 and particularly 2007 (Figures 2a and 2b). Annual precipitation was 1,247 mm, 1,472 mm, and 1,957 mm at Yosemite Valley, Wawona, and Gin Flat in 2006, respectively, compared to 568 mm, 631 mm, and 736 mm

in 2007 (Figure 2a). Annual precipitation was 1,039 mm in 2008 at Wawona. The annual





precipitation records in 2008 were incomplete at Yosemite Valley and Gin Flat. Precipitation primarily occurred from October to April or May each year, and little occurred during summer and early fall (Figure 2a).

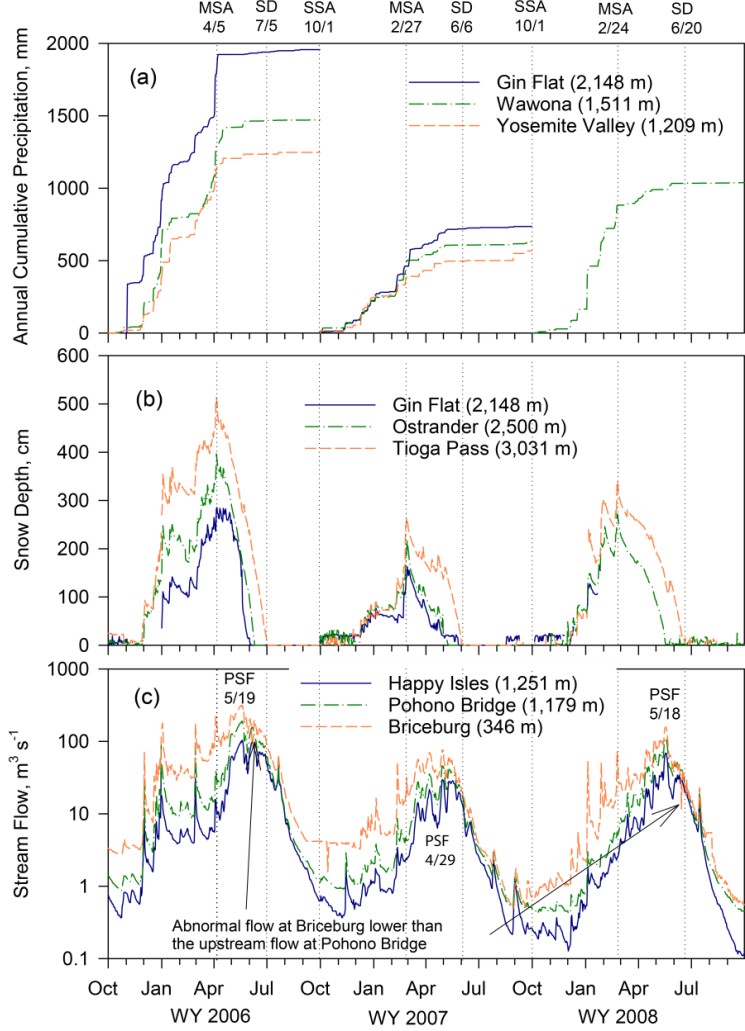

Figure 2. Hydrometeorology of the mid Merced River catchment for (a) daily accumulated precipitation, (b) daily snow depth, and (c) daily stream flow. Note the abnormal flow occasionally measured at Briceburg, which is lower than that upstream at Pohono Bridge. Also note the lack of precipitation data in 2008 at Gin Flat and Yosemite Valley and snow depth data in most of 2008 at Gin Flat and in February 2008 at Tioga Pass. The grey, dotted vertical grids mark the dates of the maximum snow accumulation (MSA), snow depletion (SD) at Tioga Pass, and start of snow accumulation (SSA) as October 1 each year; Dates of peak stream flow (PSF) were also marked in (c).



Maximum snow accumulation occurred on April 5 in 2006, with a depth of 282 cm at Gin Flat, 396 cm at Ostrander and 514 cm at Tioga Pass (Figure 2b). The snowpack was depleted at the three sites by June 5, June 11 and July 5, respectively. Maximum snow accumulation occurred

on February 27 in 2007, about 5 weeks earlier than in 2006, with maximum snow depths of 142, 192 and 264 cm at Gin Flat, Ostrander and Tioga Pass, respectively, approximately 50% of the depth in 2006. Snowpack depletion occurred in late May and early June of 2007 at all snow course sites. Snowpack reached a maximum depth on February 24 in 2008, similar to 2007, but with a much deeper snowpack (272 and 339 cm at Ostrander and Tioga Pass, respectively; Note that snow

depth data was not available for most of 2008 at Gin Flat). Snowpack was mostly depleted by late May and June in 2008 at Ostrander and Tioga Pass, respectively.

The hydrograph in the Merced River follows a typical pattern of a snowmelt-dominated hydrologic system of the U.S. West, steadily increasing in early spring, peaked in mid spring or late spring and then gradually decreasing (Figure 2c). Peak stream runoff occurred on May 19 in

2006, measured at 103 and 191 $m^3$ $s^{-1}$ at Happy Isles and Pohono Bridge, respectively. Peak flows higher than these values have been recorded only 13 times from 1916 to 2008 at the same gages. Peak flows occurred earlier in drier 2007 on April 29, with only 30 and 46 $m^3$ $s^{-1}$ at Happy Isles and Pohono Bridge, respectively. Peak flows below these values have been recorded only 11 times from 1916 to 2008. The flow condition in 2008 was intermediate, with peak flows of 69 and 112

$m^3$ $s^{-1}$ on May 18, 2008, at Happy Isles and Pohono Bridge, respectively. Several flow spikes usually occurred before the peak flow, apparently driven by rainfall events. The flows at Briceburg were occasionally lower than the upstream location at Pohono Bridge (Figure 2c), showing the occasional problems on flow measurements at Briceburg as mentioned earlier.

Based on the information above, a water year was divided into four periods to facilitate

understanding the temporal variability of isotopic composition in stream water in the following sections. Four periods were: (1) snow accumulation period from October 1 (previous year) to maximum snow accumulation (MSA) in spring at Tioga Pass; (2) snowmelt rising period from MSA to peak stream flow (PSF) at Happy Isles and Pohono Bridge; (3) snowmelt receding period from PSF to snow depletion (SD) at Tioga Pass; and (4) baseflow period from SD to September

30. Snow depletion dates at Tioga Pass were chosen in consideration of the entire mid Merced River catchment. Snow depleted several weeks earlier in lower elevations (e.g., Gin Flat) than Tioga Pass (Figure 2b). The snow depletion dates at Tioga Pass would be too late to mark the end





of snow cover for many small catchments, which are mostly located below 2,500 m - the upper limit of the snow-rain transition zone (Figure 1). However, snow at the observation sites melted

out several weeks before the basin itself was free of snow (*Rice et al*., 2011). In addition, snowpack was much deeper in higher elevations than lower elevations (Figure 2b) and the depletion of snowpack in the areas above Tioga Pass should occur much later than that at Tioga Pass. Therefore, using snow depletion dates at Tioga Pass to represent the entire mid Merced River catchment appears to be a balanced consideration following the rule of thumb.


### 4.2. Isotopic composition in precipitation, stream water and groundwater

Mean isotopic values varied significantly over locations in precipitation, stream water and groundwater and from precipitation to stream water and groundwater (Table 1). The mean $\delta^2H$ values ranged from -80.2 to -106.5‰ in snowpits excavated at the maximum snow accumulation

in spring 2006 (Dana Lake samples not included) and in precipitation collected at NADP site from November 2006 to April 2007, with an elevation range of 1,393-2,500 m. The mean $\delta^2H$ values varied from -90.5‰ to -102.4‰ in stream water along the Merced River above Briceburg and from -64.2‰ to -101.3‰ in tributaries with a mean drainage elevation ranging from 913 m to 2,969 m. The mean $\delta^2H$ values in four springs varied between -67.6‰ and -99.0‰, with sampling

locations ranging in elevation from 372 to 1,210 m, and between -80.4‰ and -94.1‰ in groundwater, with sampling ports ranging in elevation from 544 to 1,994 m.

Temporal variability of $\delta^2H$ values, as illustrated by $1\sigma$ (1 standard deviation) values in Table 1, was the greatest in snow and precipitation, with $1\sigma$ ranging from 10.5‰ to 21.0‰, and generally the lowest in spring and groundwater, with $1\sigma < 3.0‰$ for most sites. The $1\sigma$ $\delta^2H$ value

varied from 4.1‰ to 7.5‰ for stream water samples collected in the Merced River above Briceburg and < 3.6‰ for all tributaries except Yosemite Creek and the South Fork (8.8‰ and 7.0‰, respectively).

$\delta^2H$ values in snow and precipitation varied significantly between storms. $\delta^2H$ values in precipitation at NADP site in the Park ranged from -109.9‰ to -54.3‰ from November 2006 to

April 2007 at an elevation of 1,393 m (Figure 3a). $\delta^2H$ values in snowpits at much higher elevations also changed significantly over depth, with a range of -126.8 to -72.6‰ at Badger Pass (elev. 2,226 m) and -159.4 to -71.6‰ at Ostrander (elev. 2,500 m) (Figure 3b and 3c). It was impossible in this





study to associate the variation of $\delta^2H$ values over snow depth with storm history but nevertheless

it approximately reflected the temporal changes of $\delta^2H$ values in snowfall over time.


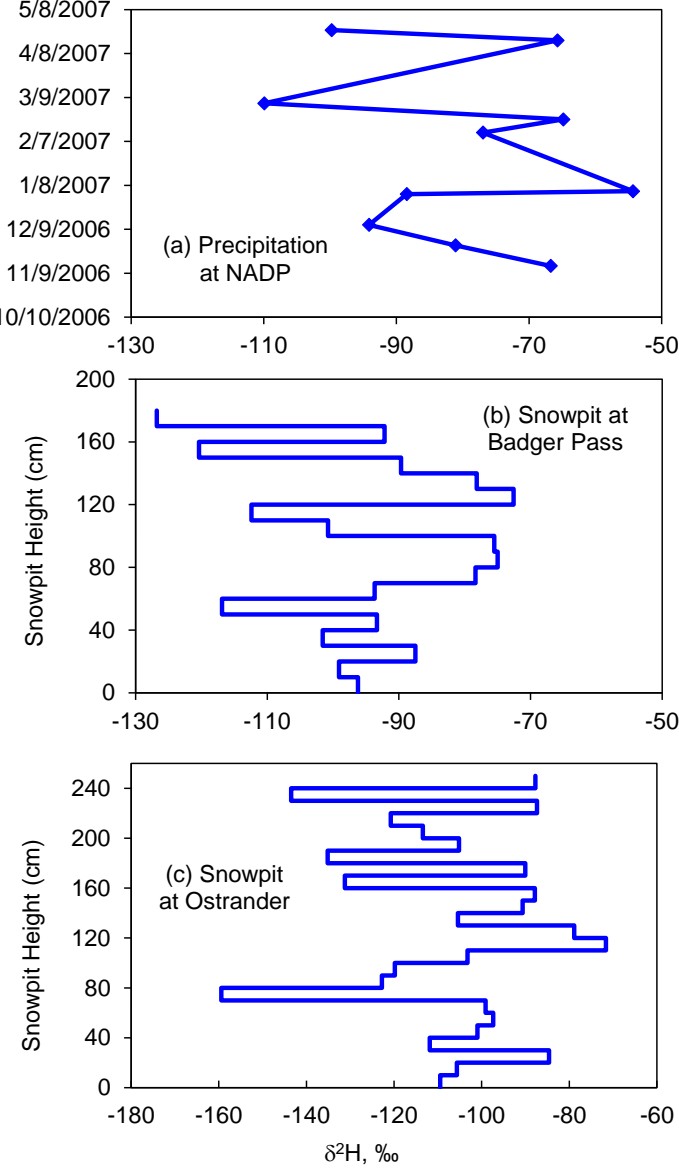

Figure 3. (a) Temporal variation of $\delta^2H$ in precipitation at the National Atmospheric Deposition Program (NADP) site located in Yosemite Valley; (b) and (c) $\delta^2H$ profiles in snowpits excavated at the maximum snow accumulation at Badger Pass and Ostrander, respectively.

$\delta^2$H values in stream water along the Merced River varied over time, with more depleted (lower) values during the snowmelt period (snowmelt rising + receding periods) and more enriched values (higher) during the snow accumulation and baseflow periods (Figure 4). $\delta^2$H values in stream water along the Merced River became more enriched with an increase in drainage areas or a decrease in sampling elevations, with the lowest values at Happy Isles and the highest values at

Briceburg consistently from 2006 to 2008 except for a couple of samples.

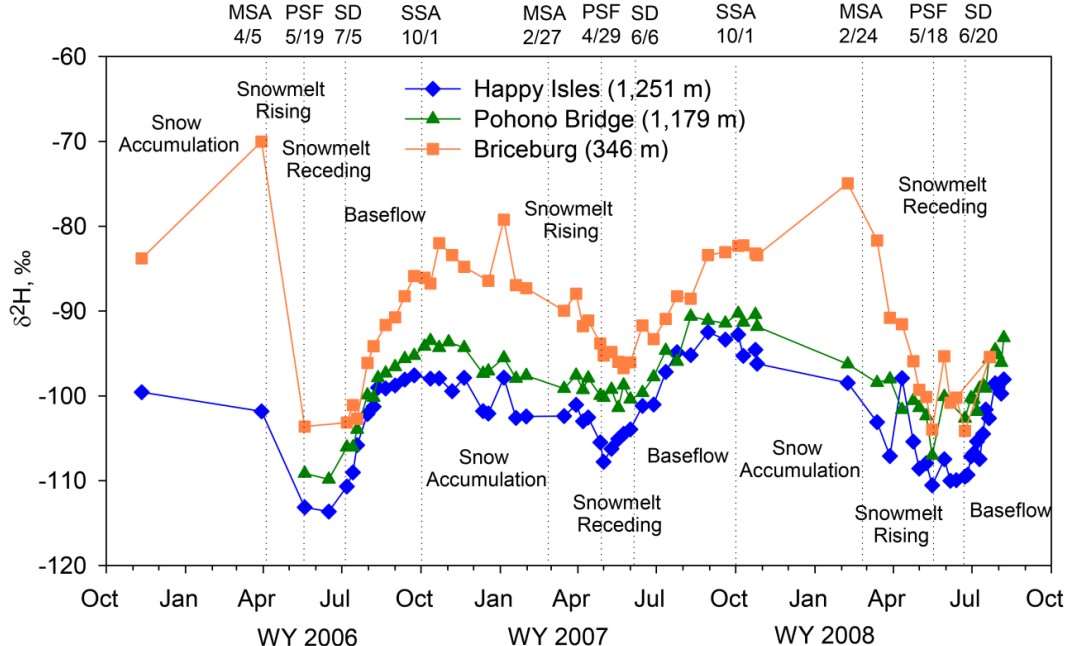

Figure 4. Variation of $\delta^2$H values in stream water from water years 2006 to 2008 at Happy Isles, Pohono Bridge and Briceburg. Dates marked by gray, dotted vertical grids are the same as in Figure 2 with addition

of peak stream flow (PSF). Four periods were also marked wherever space is allowed.

During the snow accumulation period, isotopic composition in the Merced River tended to become gradually depleted at Happy Isles, Pohono Bridge and Briceburg (Figure 4). For example, $\delta^2$H values were -98.0‰ on October 12, 2006, and -102.4‰ on January 31, 2007, at Happy Isles.

There were isolated spikes in isotopic values during the period, e.g., a spike on January 5, 2007, at all three gages and on February 8, 2008, at Briceburg. These isolated spikes appear to be caused by rain events with more enriched isotopic composition. For example, a major rain event occurred



on January 4, 2007, with 12 mm recorded at Yosemite Valley and $\delta^2$H value of -54.3‰ at NADP
site (Figure 3a), which increased stream flow and $\delta^2$H values in stream water abruptly the next day

at all three gages (Figures 2c and 4). During this period, $\delta^2$H values decreased with an increase in
stream flow by a logarithmic function at Happy Isles and Pohono Bridge, but increased at
Briceburg (Figure 5). Though the positive relationship at Briceburg was not significant ($p > 0.05$),
it was apparently a result of greater rainwater inputs with more enriched isotopic signature. The
magnitude of stream flow spikes at Briceburg was much higher than at the other higher elevation

gages during the snow accumulation periods, suggesting much more rainfall inputs from lower
elevations at Briceburg (Figure 2c), causing an increase in isotopic values in stream water with an
increase in stream flow.

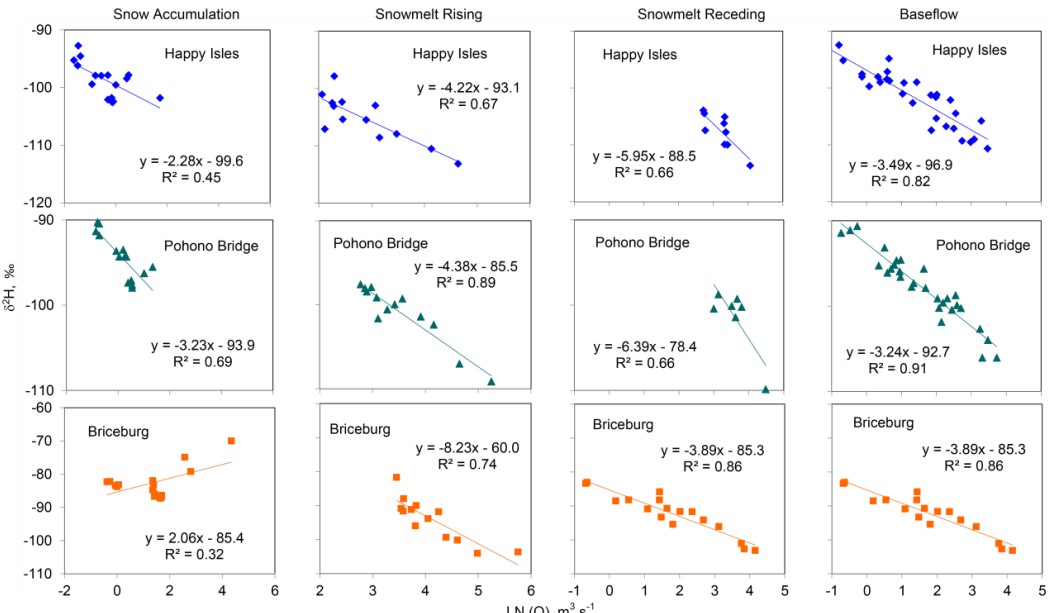

Figure 5. Correlation between $\delta^2$H values in stream water and stream flow (natural logarithmic values)
  during four periods at Happy Isles, Pohono Bridge and Briceburg.

   During the snowmelt period (snowmelt rising + receding periods), the variation of $\delta^2$H
  values over time followed the shape of a trough (Figure 4). In fact, the variation can be described

by a parabola function, particularly for 2006 and 2008 at Happy Isles ($R^2 = 0.98$ and 0.91,
  respectively; curves not shown). The lowest values, which occurred at peak flows, were





significantly inversely correlated with peak flows ($R^2 = 1.0$, $n = 3$, $p < 0.05$) and varied over years, e.g., -113.7‰ in 2006, -107.8‰ in 2007 and -110.6‰ in 2008 at Happy Isles. During the period, isotopic composition became depleted with an increase in stream flow ($p < 0.05$), consistent

between the snowmelt rising and receding periods for all three Merced River gages (Figure 5).

During the baseflow period, isotopic composition became enriched over time (Figure 4). The isotopic enrichment over time during this period occurred much more rapidly (steeper slopes) than the isotopic depletion during the snow accumulation period. Also, the enrichment was much stronger at Briceburg (again steeper slopes) than at Happy Isles and Pohono Bridge, particularly

in 2006 and 2007. During the period, $\delta^2H$ values decreased with an increase in stream flow significantly ($p < 0.05$) at all three Merced River gages (Figure 5).

### 4.3. Local meteoric water line and local evaporation line in stream water and groundwater

A local meteoric water line (LMWL) of $\delta^2H$ versus $\delta^{18}O$ was established using 71 snow

and rain samples collected at NADP site and snowpits (each 10-cm snow sample treated as an individual sample for this purpose) excavated at Badger Pass, Gin Flat, and Ostrander (Figure 6a). The slope and intercept of the LMWL were 7.88 and 9.39 ($R^2 = 0.96$, $p < 0.001$), respectively, which are very close to those (8 and 10, respectively) of the global meteoric water line (GMWL) of *Craig* (1961).

Most stream water samples collected along the Merced River and its tributaries fall near LMWL on the $\delta^2H$-$\delta^{18}O$ plot (Figures 6b and 6c). However, the slopes of $\delta^2H$-$\delta^{18}O$ linear trends for individual sites were lower than that of LMWL and varied over locations (Table 2), indicating an evaporation effect. The slope was lower than 6.13 for all Merced River locations, with the intercept less than -14.7. For tributaries, the slope and intercept were even lower, e.g., slope < 5.0

in seven of eight tributaries and intercept mostly less than -30.0 (Table 2). $R^2$ values varied from 0.73 to 0.90 for all Merced River locations except Cascade Picnic Area (0.48), but were lower than 0.76 for all tributaries except Yosemite Creek (0.95) and the South Fork (0.94).

Almost all Merced River samples collected during the snow accumulation period are located right below the LMWL (Figure 6b), showing a local evaporation line (LEL) with a slope

of 7.29 and an intercept of -0.72 ($n = 81$, $R^2 = 0.93$) (Table 2). Merced River samples collected during the snowmelt rising period are scattered near LMWL except for one outlier on lower left of LMWL (Figure 6b), with a slope of 6.08 and an intercept of -15.34 ($n = 75$, $R^2 = 0.77$) for LEL



(Table 2). During the snowmelt receding period, most samples were below LMWL (Figure 6b) and the slope and intercept of LEL were 6.61 and –9.19 ($n = 50$, $R^2 = 0.73$), respectively (Table

2). During the baseflow period, all samples other than a few were below LMWL (Figure 6b) and the slope and intercept of LEL were 6.00 and –18.58 ($n = 134$, $R^2 = 0.89$), respectively (Table 2). The samples, highlighted in an orange rectangle box on Figure 6b, were further away from LMWL and collected in the Merced River at Briceburg and the South Fork confluence during the baseflow period.


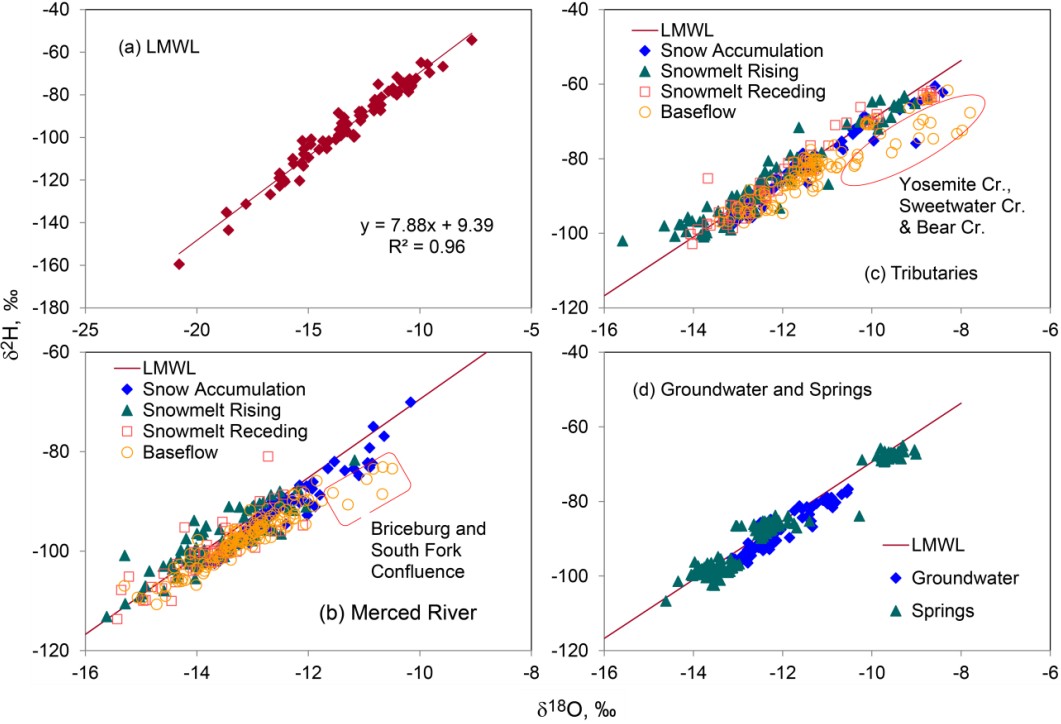

Figure 6. Relationship between $\delta^2H$ and $\delta^{18}O$ values in (a) precipitation (rain and snow); and stream water samples collected during four periods defined in Figure 4 for (b) Merced River at all locations listed in Table 1; (c) all tributaries listed in Table 1; and (d) groundwater and spring water collected at all sites.


Compared to the Merced River, the result of tributaries by periods was somewhat different. Other than the baseflow period (particularly those circled by an orange oval), samples are scattered more closely around LMWL during all periods (Figure 6c). The slope of LEL was greater than 7.0 and noticeably higher than those of the Merced River (Table 2). The intercept was also higher,



ranging from -0.19 to 3.52 and $R^2$ values were higher than 0.92. During the baseflow period, the
slope and intercept were significantly lower, 5.85 and -17.29, respectively, with an $R^2$ of 0.83,
which was primarily attributed to samples collected in Yosemite Creek, Sweetwater Creek and
Bear Creek at extremely low flows (circled in Figure 6c). It is the baseflow samples that caused
the lower slopes for individual catchments than those during the other three periods (Table 2).


Table 2. Local meteoric water line (LMWL), local evaporation line (LEL), and isotopic composition at the
intersection of LEL and LMWL.

| | Sample Number | Mean Catchment Elevation (m) | Local Evaporation Line | | | Intersection of LEL & LMWL | |
|---|---|---|---|---|---|---|---|
| | | | Slope | Intercept | $R^2$ | $\delta^{18}O$ (‰) | $\delta^2H$ (‰) |
| Precipitation for LMWL | 71 | | 7.88 | 9.39 | 0.96 | | |
| *Merced River by Catchment* | | | | | | | |
| Happy Isles | 68 | 2743 | 5.64 | -24.31 | 0.90 | -15.0 | -109.0 |
| El Capitan | 49 | 2624 | 5.51 | -25.07 | 0.89 | -14.5 | -105.0 |
| Pohono Bridge | 64 | 2580 | 5.69 | -22.11 | 0.86 | -14.4 | -103.9 |
| Cascade Picnic Area | 37 | 2539 | 4.27 | -37.75 | 0.48 | -13.0 | -93.4 |
| El Portal | 35 | 2483 | 4.94 | -31.63 | 0.73 | -13.9 | -100.5 |
| South Fork Confluence | 33 | 2350 | 4.56 | -34.51 | 0.78 | -13.2 | -94.8 |
| Briceburg | 54 | 2067 | 6.13 | -14.70 | 0.84 | -13.7 | -98.7 |
| *Merced River by Period (Samples from all catchments together)* | | | | | | | |
| Snow Accumulation | 81 | | 7.29 | -0.72 | 0.93 | -17.0 | -125.0 |
| Snowmelt Rising | 75 | | 6.08 | -15.34 | 0.77 | -13.7 | -98.7 |
| Snowmelt Receding | 50 | | 6.61 | -9.19 | 0.73 | -14.6 | -105.6 |
| Baseflow | 134 | | 6.00 | -18.58 | 0.89 | -14.9 | -107.7 |
| *Tributaries by Catchment* | | | | | | | |
| Tenaya Creek | 43 | 2528 | 3.20 | -53.93 | 0.57 | -13.5 | -97.3 |
| Yosemite Creek | 50 | 2516 | 4.67 | -33.35 | 0.95 | -13.3 | -95.5 |
| Bridalveil Creek | 48 | 2232 | 4.31 | -35.18 | 0.76 | -12.5 | -88.9 |
| Cascade Creek | 38 | 2228 | 4.95 | -25.52 | 0.61 | -11.9 | -84.4 |
| Crane Creek | 37 | 1621 | 3.92 | -34.94 | 0.75 | -11.2 | -78.8 |
| South Fork | 40 | 1857 | 5.56 | -19.42 | 0.94 | -12.4 | -88.2 |
| Sweetwater Creek | 32 | 1058 | 1.95 | -50.49 | 0.24 | -10.1 | -70.2 |
| Bear Creek | 29 | 913 | 3.40 | -33.47 | 0.61 | -9.6 | -66.0 |
| *Tributaries by Period (Samples from all catchments together)* | | | | | | | |
| Snow Accumulation | 71 | | 7.47 | 3.52 | 0.93 | -14.2 | -102.6 |
| Snowmelt Rising | 82 | | 7.01 | -0.19 | 0.92 | -11.0 | -77.1 |
| Snowmelt Receding | 59 | | 7.32 | 2.47 | 0.94 | -12.3 | -87.5 |
| Baseflow | 105 | | 5.85 | -17.29 | 0.83 | -13.1 | -94.1 |
| Springs (All) | 148 | | 7.55 | 4.76 | 0.95 | -13.8 | -99.8 |
| Groundwater (All) | 59 | | 7.22 | -0.83 | 0.86 | -15.3 | -111.2 |

Note that the last four tributaries listed in Table 1 were not inlcuded here because their $\delta^2H$-$\delta^{18}O$ relationship was
not significnat ($p > 0.05$) due to the lack of samples. Also, see text for discussion about the division of four
periods for a water year.





The $\delta^2$H-$\delta^{18}$O relation in groundwater and springs were closer to LMWL than in stream

water (Figure 6d and Table 2). The slope and intercept of the evaporation lines were 7.22 and -

0.83 for groundwater and 7.55 and 4.76 for spring water, respectively.

**4.4. Variation of isotopic values in stream water, groundwater and precipitation with elevation**

Mean isotopic values of stream water from relatively small catchments (8-122 km$^2$; including all listed under tributaries in Table 1 except the South Fork), groundwater, and rock glacier outflows were highly correlated with mean elevations of their catchment areas (Figure 7a and 7b). The slope and intercept were -0.0022 and -7.57 for $\delta^{18}$O (R$^2$ = 0.91, $n$ = 16, $p$ < 0.001), respectively, and -0.019 and -48.7 for $\delta^2$H (R$^2$ = 0.96, $n$ = 16, $p$ < 0.001). The Crane Flat and

Hodgdon Meadow wells are located near the mid Merced River divide (inside and outside, respectively) and far away from major streams (Figure 1). Groundwater in these wells was deemed to be derived from precipitation in the drainage area above each well. These drainage areas, along with the mean drainage elevations, were computed the same as for a stream sampling location using well locations as pour points. The result indicates that elevations vary narrowly from the

well locations to the drainage summit at Crane Flat and Hodgdon Meadow, with a relief of only 33 and 429 m, respectively (Table 1). A similar analysis cannot be performed for the other groundwater wells due to the complex topography and their proximity with the Merced River and thus samples from those wells were excluded in this analysis.

Variation of isotopic values in snow with sampling elevation was examined using mean

isotopic values from four snowpits excavated along an elevation gradient and a rain gage located at Yosemite Valley (Figure 7a and 7b). The slope of the $\delta^2$H-elevation linear relationship was identical to that of small streams, groundwater, and rock glacier outflows and the intercept was also very close (-51.3 versus -48.7), even though its R$^2$ value was much lower (R$^2$ = 0.71, $n$ = 5, $p$ = 0.07).

An analysis was also conducted to exclude samples of two groundwater wells and three rock glacier outflows outside the mid Merced River catchment (Figure 7c). The result indicated that the $\delta^2$H-elevation relationship did not change significantly, with a slope of -0.016 and intercept of -52.5 (R$^2$ = 0.94, $n$ = 11, $p$ < 0.001).

To examine if evaporation affected the isotope-elevation relationship, the mean isotopic

values in stream water were corrected using both LMWL and LEL (Table 2). Using the isotopic values at the intersection between LMWL and LEL, the isotope-elevation relationship was still significant for small streams ($R^2$ = 0.96 for $\delta^2H$, $n$ = 7, $p$ < 0.001) and yielded a similar slope (-0.017) and intercept (-50.5) (Figure 7d).

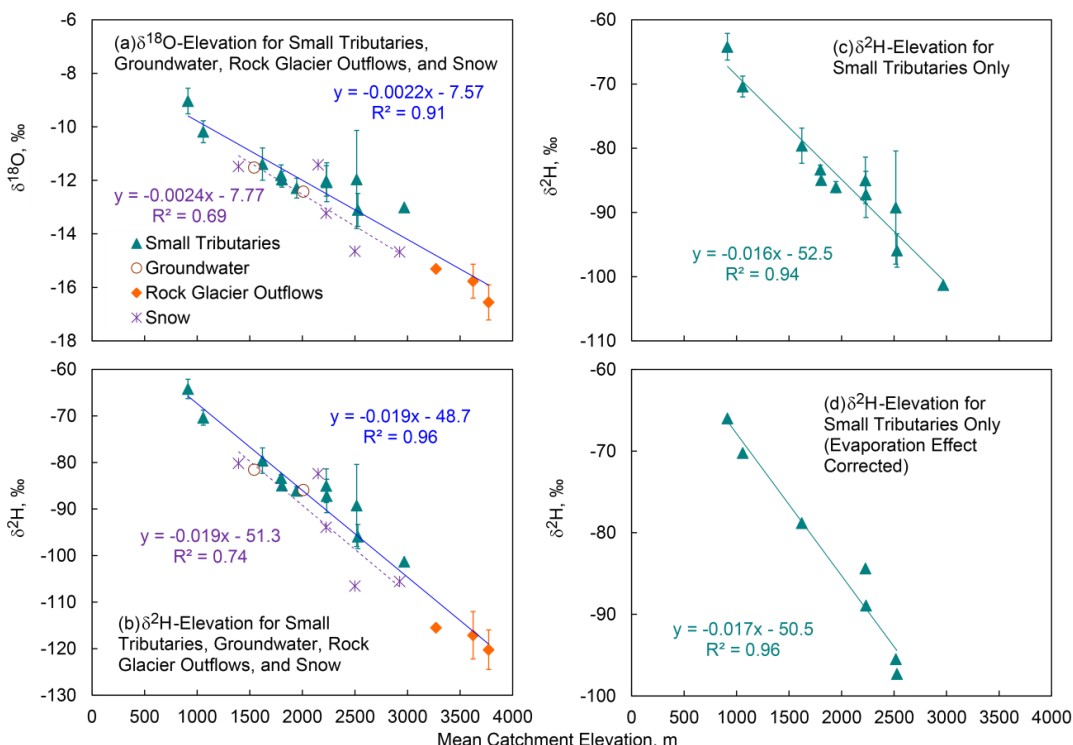


Figure 7. Variation of isotopic composition with mean catchment elevations: (a) and (b) for $\delta^{18}O$ and $\delta^2H$ values, respectively, in small tributaries (catchment area < 122 km$^2$), groundwater with estimated source water elevations (Crane Flat and Hodgdon Meadow), and rock glacier outflows, along with snow and rain samples. The blue solid line shows the linear trend for small tributaries, groundwater and rock glacier

outflows and the dashed purple line for snow and rain. (c) for $\delta^2H$ values in small tributaries without groundwater and rock glacier outflows; and (d) for $\delta^2H$ values in small tributaries with evaporation effect corrected by local meteoric water line. The number of samples in (d) is less than in (c) due to the lack of samples to establish a significant relationship between $\delta^2H$ and $\delta^{18}O$ values for the last four tributaries listed in Table 1.






Seasonal variation of the $\delta^2$H-elevation relationship was examined using samples collected during the four periods defined earlier (Figure 8). The slopes and intercepts of $\delta^2$H-elevation linear relationship did vary over the periods, but not remarkably. The slope varied between -0.015 and -0.021 and the intercept values between -40.3 and -55.0 for all these periods except the snow

accumulation period and the snowmelt rising period in 2006. Samples were not collected in tributaries in spring and summer of 2006 and the samples collected in the snow accumulation period in 2006 did not cover a wide range of elevations. The slope and intercept did not appear to change significantly from the snowmelt rising period to the snowmelt receding period in 2007 and 2008. Merced River samples were also plotted independently in Figure 8. It is apparent that Merced

River samples did closely follow the trend of small tributaries.

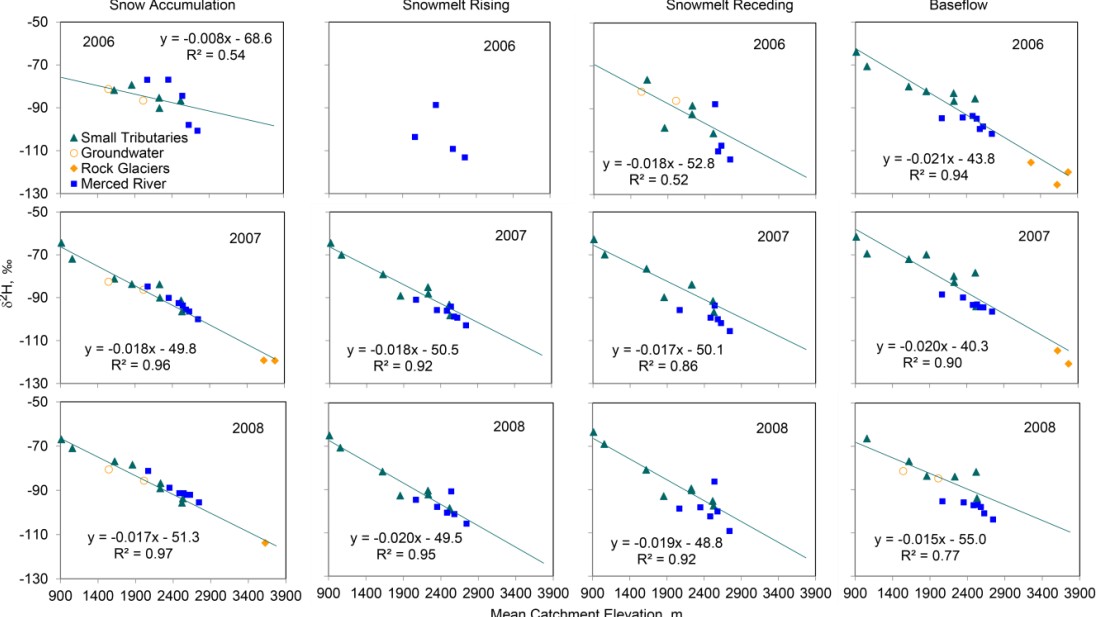

Figure 8. Seasonal variation of the $\delta^2$H-elevation relationship in small tributaries, groundwater and rock glacier outflows, with a linear trend (green). Samples from Merced River were also plotted, but not included

in establishing the trend line. Four periods were defined the same as Figure 4. The number of samples for each analysis varies depending on the availability of samples. Note that no samples were available for tributaries, groundwater and rock glacier outflows during the snowmelt rising period in 2006 due to the road blockage caused by a massive land slide.



## 5. Discussion and application

### 5.1. Controls on isotopic composition in stream water and groundwater

5.1.1. Elevation effect

Elevation exerts a major control on the mean isotopic values in stream water at small catchments (including rock glacier outflows) and groundwater in the mid Merced River catchment (Figures 7a and 7b), which is consistent with *Jeelani et al.* (2010). Unlike monsoon precipitation samples collected along an elevation gradient in India (*Kumar et al.*, 2010), the slopes and intercepts of their correlations did not vary much over seasons and years with dramatically different hydrologic conditions (Figure 8). The elevation gradient determined by those samples, e.g., -0.22‰/100m for $\delta^{18}$O and -1.9‰/100m for $\delta^{2}$H on average (Figures 7a and 7b), essentially represents lapse rate of isotopic composition in meteoric water in the mid Merced River catchment. This lapse rate is caused by Rayleigh distillation as the heavier isotopes are concentrated in the precipitation, resulting in clouds progressively becoming isotopically lighter with ascending to higher elevations or moving further away from ocean (*Poage and Chamberlain*, 2001; *Clark and Fritz*, 1997). The mean lapse rate of this study is reasonably close to that obtained elsewhere around the world, which averaged -0.28‰/100 m for $\delta^{18}$O (*Poage and Chamberlain,* 2001). The lapse rate of $\delta^{18}$O is identical to that of precipitation in a south Ecuadorian montane cloud forest catchment (San Francisco catchment, 1,800-2,800 m) (*Windhorst et al.*, 2013) and almost the same as that of precipitation in the upper Heihe River in the northwestern China (1,674-5,103 m), where a gradient of -0.18‰/100m was obtained (*Wang et al.*, 2009b). It is also very close to the gradient in northern California, where $\delta^{2}$H values in groundwater changed from -40‰ to -120‰ from the coast to the Sierra crest with a relief of 4,000 m, with a lapse rate of -2.0‰/100m (*Ingraham and Taylor*, 1991). Since the isotopic lapse rate did not change longitudinally in Sierra Nevada (*Friedman and Smith*, 1970), this lapse rate may be applicable to the western slope of the entire Sierra Nevada.

However, this lapse rate is significantly lower than that (-4‰/100 m for $\delta^{2}$H) reported earlier by *Friedman and Smith* (1970) using snow-core samples collected around April 1 of 1969 in the west slope of Sierra Nevada. The lapse rate of *Friedman and Smith* (1970) also does not agree with the result of our snow samples (Figures 7a and 7b). The discrepancy in the results between our snow samples and those of *Friedman and Smith* (1970) is primarily caused by significant temporal variability of isotopic composition in snowpack over seasons and years and





uneven temporal variation over elevation bands as found by *Jodar et al.* (2016) for the European Alps. For example, $\delta^2$H value in a snowpit at Gin Flat (elevation = 2,150 m) was -103‰ reported by *Friedman and Smith* (1970) but -81.5‰ in this study, with a difference of 21.5‰. $\delta^2$H value was -139‰ at Big Whitney Meadow (elevation = 2,970 m) in 1969, whereas it was -105.5‰ at

similar elevation (2,926 m) at Dana Lake in 2006, with even a greater difference than at Gin Flat at 33.5‰. It was very wet in 1969, with annual precipitation of 1,649 mm compared to 1,247 mm in 2006 at Yosemite Valley. Information on snowfall amount or snow depth in 1969 was not available, but heavier storms usually result in lighter stable isotopes in snow (*Ingraham*, 1998). In addition, snow is usually subject to isotopic fractionation if sublimation and melting occur (*Taylor*

*et al.*, 2001; *Earman et al.*, 2006; *Frisbee et al.*, 2009; *Earman et al.*, 1996). *Dettinger et al.* (2004) demonstrated that melting and sublimation did occur in the snowpack in Sierra Nevada before April 1. It is not possible to evaluate how significant isotopic fractionation has affected the isotopic composition in the snow samples collected by *Friedman and Smith* (1970), as $\delta^{18}$O was not analyzed in their study. However, the isotopic composition in the snow samples of this study,

which was mostly collected at the maximum accumulation, was very close to GMWL of *Craig* (1961) (Figure 6a), indicating that isotopic fractionation effect due to sublimation was not evident in our snow samples.

      Using samples from precipitation, the lapse rate may vary significantly over years and seasons and is not always reliable (*Hemmerle et al.*, 2021). *Gamboa et al.* (2022) demonstrated

that the lapse rate of $\delta^2$H varied from -1.4 to -3.5‰/100m using precipitation samples collected during intermittent periods from 1984 to 2017 in the Atacama Desert of the Northern Chile. From the same study, the lapse rate of $\delta^2$H was -1.6‰/100m using groundwater samples and the mean sub-basin elevations, which is very close to ours. Furthermore, the lapse rate may vary dramatically with different climates, particularly when precipitation samples are used. For example, the lapse

rate of $\delta^2$H was -0.8‰/100m (summer) and -0.9‰/100m (winter) in the arid and semi-arid Tucson Basin in the Southern Basin-and-Range Province of Arizona and New Mexico (*Eastoe and Wright*, 2019), -0.7‰/100m in the humid Great Lakes region in the Eastern Democratic Republic of the Congo (*Balagizi et al.*, 2018), and -3.4‰/100m in the Juncal River basin of Central Chile (2,200-3,000m) (*Ohlanders et al.*, 2013).






### 5.1.2. Evaporation effect

All samples of the Merced River, tributaries, groundwater and spring water were very close to LMWL in the $\delta^2$H-$\delta^{18}$O bivariate plots exempt for some collected during the baseflow period
(Figure 6). The slopes of the local evaporation lines in groundwater and spring water were only slightly lower than that of LMWL (Table 2), indicating that evaporation during groundwater recharge was not very strong. However, the slopes of LEL in the Merced River and tributaries were noticeably lower than that of LMWL (Table 2), showing an apparent evaporation effect, the same as *Jeelani et al.* (2013) and *Reckerth et al.* (2017).

Both the slope and $R^2$ values of LEL were generally lower in tributaries than in the Merced River except for $R^2$ values at Yosemite Creek and the South Fork when LELs were constructed using data from individual catchments (Table 2). The lower slopes in tributaries were primarily caused by samples collected during low flows in later summer and fall, particularly those with waterfalls such as Yosemite Creek and wider but shallower channels such as the South Fork
(Figure 6c). When all samples were grouped into four periods, the slopes and $R^2$ values of LEL in tributaries became much higher and closer to LMWL than those in the Merced River during all periods other than the baseflow period (Table 2). Apparently, evaporation was stronger in the Merced River than in tributaries during all periods other than the baseflow period. During the baseflow period, stronger evaporation occurred in tributaries, particularly in Yosemite Creek
(Figure 6c). However, the isotope-elevation relation established using small tributaries and groundwater was not strongly affected by evaporation and the isotopic composition in the Merced River was still primarily controlled by source waters from various elevations even during the baseflow period (Figure 8).

### 5.1.3. Snowmelt and isotopic fractionation effects

The temporal variability of isotopic values in snow was much higher than that of stream water (Figures 3 and 4; Table 1). Isotopic composition in stream water over three water years with very different precipitation amounts has attenuated much of the temporal variability of stable isotopes in precipitation, consistent with the observation of *Kendall and Coplen* (2001), *Dutton et*
*al.* (2005), *Jeelani et al.* (2013), and *Reckerth et al.* (2017). The variability attenuation primarily explains why the isotope-elevation relations did not vary dramatically when stream samples were used (Figure 8). Compared to the variability of isotopic composition in groundwater and spring





water, however, the isotopic composition in stream water still varied significantly over seasons (with respect to 1σ values in Table 1). During snowmelt, $\delta^2H$ values in stream water at Happy Isles, Pohono Bridge and Briceburg were much lower than during the other periods (Figure 4). This result was apparently caused by the snowmelt contribution to streams from melting snowpack, supported by *Shaw et al.* (2014) and *Liu et al.* (2017). However, the seasonality did not significantly change the slopes of $\delta^2H$-elevation relationship over seasons (Figure 8). Also, $\delta^2H$ values in stream water were consistently distinct from 2006 to 2008 over sampling locations at Happy Isles, Pohono Bridge and Briceburg, except for a few samples that were affected by rainfall events (Figure 4). It is suggested that even during snowmelt, elevation still exerts a major control on the isotopic composition in stream water in the mid Merced River catchment.

Studies have shown that snowmelt becomes isotopically enriched over time due to isotopic fractionation between ice and liquid water (e.g., *Taylor et al.*, 2001; *Earman et al.*, 2006). As a result, isotopic values in snowmelt from a snowmelt lysimeter were significantly lower than those in the bulk snowpack before the peak snowmelt and higher after that, resulting in a monotonic curve with isotopic values gradually increasing over time in snowmelt and stream water (*Liu et al.*, 2004). The variation of $\delta^2H$ values during the snowmelt period in the Merced River followed a parabola curve (the curve not shown but the trend can be seen in Figure 4), instead of a monotonic one. In addition, the difference between the snowmelt rising and receding periods was not evident for $\delta^2H$-flow relationship, $\delta^2H$-$\delta^{18}O$ relationship, and $\delta^2H$-elevation relationship (Figures 5, 6, and 8). These results suggest that isotopic fractionation between ice and liquid water in snowmelt did not appear to affect much the isotopic signature of stream water at the catchment scales involved in this study.

### 5.2. Applications and implications

The lapse rate of stable isotopes (or the isotope-elevation relation) in meteoric water acquired by this study would be useful for paleoelevation studies as demonstrated for Sierra Nevada of California by *Mulch et al.* (2006). This information is also very useful for understanding source waters (e.g., *Jean-Baptiste et al.*, 2022; *Jeelani et al.*, 2013) and the sensitivity of stream flow in response to climate change. For the latter, for example, stream flow during the baseflow period at lower elevations (e.g., Briceburg of this study) is more strongly affected by rainfall and thus more sensitive to changes in snow-rain ratio in the future as alluded by Figure 5 and the





relevant text in section 4.2. Below are two additional examples of its applications in watershed
hydrology and hydrometeorology.

### 5.2.1. Building conceptual understanding on hydrometeorologic processes

Based on the discussion in section 5.1, a catchment characteristic isotopic value (CCIV) of
source waters – isotopic composition at the mean catchment elevation that represents source waters
from the entire catchment - can be defined by the isotope-elevation relation for all sub-catchments
in the mid Merced River catchment (Figure 9). In combination with local meteoric water line,
CCIV helps elucidate hydrometeorologic processes over seasons. In the Merced River at Happy
Isles, for example, $\delta^2H$ value was below CCIV starting on March 30, 2006, and near CCIV again
on August 7, 2006, after a trough-shape turn (Figure 9a). These two dates approximately match
the start and end of the snowmelt season for 2006 based on stream flow. The start date was also
very close to the maximum snow accumulation date (Figures 2b and 9a). The end date was about
four weeks later than the snow depletion date at Tioga Pass. Based on *Rice et al.* (2011), snow at
the observation sites melted out several weeks before the catchment itself was free of snow.
Therefore, the end date also appears to match the end of snowmelt. The snowmelt duration
determined this way in 2007 and 2008 also agrees reasonably well with that determined by stream
flow. Similarly, the results from Pohono Bridge and Briceburg (not shown) are consistent with
Happy Isles. The intersection of CCIV line and the isotopic time series curve marks reasonably
well the snowmelt duration. Since isotopic values are highly correlated with stream flows (Figure
5), in addition, the lowest isotopic value during the snowmelt period can be used to infer the
relative magnitude of snowmelt event. The lower the isotopic value at the bottom of trough the
higher the magnitude of snowmelt event. This approach may be used to determine snowmelt
duration and relative magnitude for ungagged basins without stream flow measurements.

In the Merced River at Happy Isles, $\delta^2H$ values were above CCIV line during the baseflow
periods and below the line during the snow accumulation periods (Figure 9a), reflecting the shift
of source water elevations, evaporation and occasional rainfall effects as discussed earlier. The
local meteoric water line and evaporation line of groundwater could be used to assist differentiation
of the dominant processes during these periods. For example, $\delta^2H$ values were 5-8‰ more
enriched during the baseflow period in 2007 than in 2006 (Figure 9a). The enrichment for these
samples is deemed to be primarily caused by evaporation, rather than shift in source water





elevation. These samples collected in 2007 are located below and further to the right of LMWL
       and LEL of groundwater than the samples collected in 2006 (Figure 10a), indicating a stronger
       evaporation effect. Though the shift in source water elevation and evaporation cannot be
       quantitatively determined, the CCIV line helps build a conceptual understanding of
       hydrometeorologic processes.


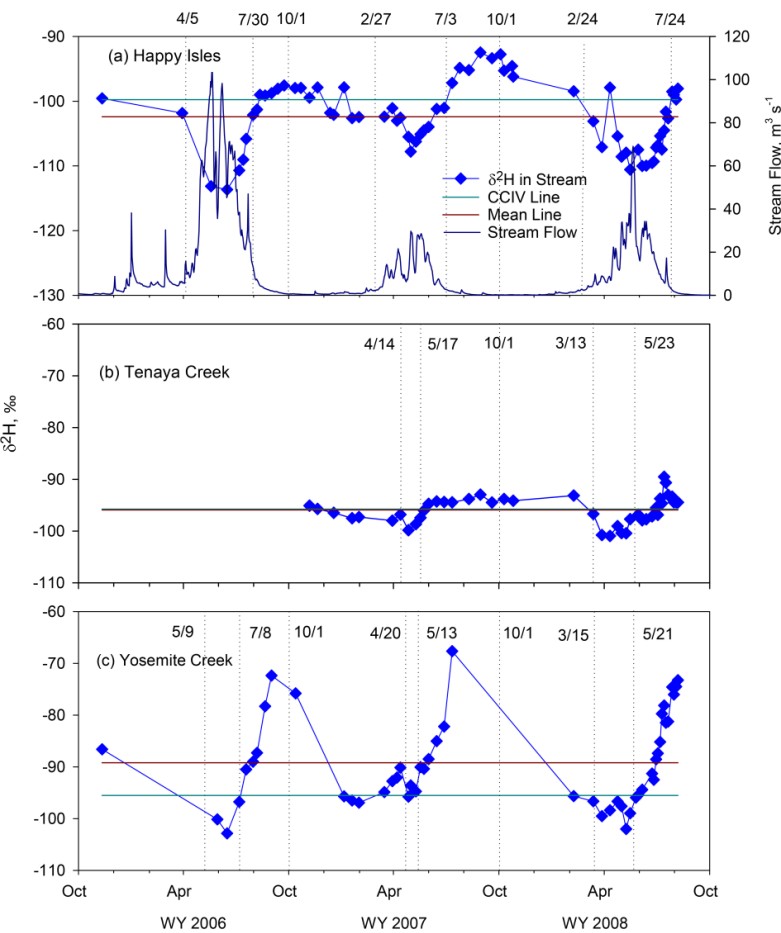

Figure 9. $\delta^2H$ values in stream water in (a) Merced River at Happy Isles, with stream flow, (b) Tenaya
Creek, and (c) Yosemite Creek, along with catchment characteristic isotopic value (CCIV) of $\delta^2H$ and a
line determined by mean $\delta^2H$ value in samples. Dates in (a) mark the start and end of snowmelt season
determined by hydrograph at Happy Isles and October 1; dates on (b) and (c) mark the start and end of
       snowmelt season using the intersections of the time series curve and CCIV. Note that the mean and CCIV
       lines overlap in (b).



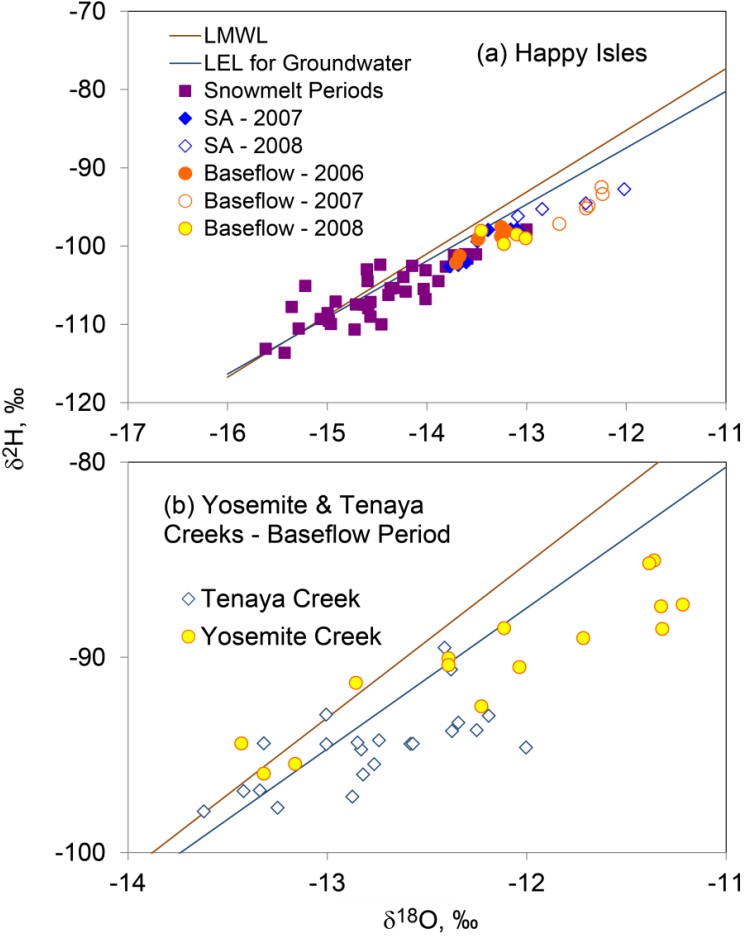

Figure 10. Scatter plot of δ²H vs. δ¹⁸O for (a) Merced River at Happy Isles during different periods and (b) comparison between Tenaya Creek and Yosemite Creek during the baseflow period. Local meteoric water line (LMWL) and evaporation line (LEL) of groundwater are also shown as references.

Comparing the temporal variation of δ²H values relative to the CCIV line between Yosemite Creek and Tenaya Creek, two ungauged streams, reveals more interesting results (Figures 9b and 9c). The two adjacent basins share many similarities, e.g., basin area, elevation ranges, and mean basin elevations (Table 1 and Figure 1), other than Yosemite Creek terminating with two cascading waterfalls (739 m tall) in Yosemite Valley. Indeed, the lowest δ²H values were close and occurred at about the same time, indicating that peak snowmelt occurred with similar magnitudes at about the same time in these catchments. The dates when the CCIV line and time



series curve intersected were similar, suggesting that the duration of snowmelt events appear to be close as well. However, the variation of $\delta^2$H values relative to the CCIV line was very different, with most samples particularly those collected in the baseflow periods far above the line in Yosemite Creek (Figures 9b and 9c). Compared to Tenaya Creek, the samples collected during baseflow in Yosemite Creek were plotted below and further right to LMWL and LEL of

groundwater (Figure 10b). This result indicates that evaporation was much stronger in Yosemite Creek than in Tenaya Creek and shifting of source water toward lower elevations was not the main reason. It is suggested that Yosemite Creek is much more sensitive to climate warming than Tenaya Creek. Flow in Yosemite Creek was intermittent in drier years (e.g., it dried up starting mid-July in 2007). Without even considering any effect of other factors (e.g., shift in snow-rain ratio and

the earlier onset of snowmelt), an increase in air temperature alone would increase evaporation, reduce flow, and further shorten the duration of flow in Yosemite Creek. This trend is certainly not good news for Yosemite National Park tourists as Yosemite Falls are one of the most attractive features in the park.

     One would argue that a simple horizontal line using the mean isotopic value from samples

collected in the same catchment could serve the same purpose as the CCIV line. The mean line could work if the number of samples was large enough and evaporation was known to be neglectable *a priori* such as Tenaya Creek (Figure 9b). However, it would not work for catchments with strong evaporation such as Yosemite Creek. If the mean line is applied to Yosemite Creek, it will be very misleading. The snowmelt events will be much exaggerated and evaporation effect

will be greatly under-stated.

     Based on the above analysis, a guideline is developed to identify hydrometeorologic processes using the time series of stable isotopes and the CCIV line for the mid Merced River catchment, which we think applicable to other snowmelt-fed catchments. If isotopic values in stream water are on or near the CCIV line, it indicates that source waters of stream flow are likely

from all elevations, with an approximately equal discharge rate from higher and lower elevations. If the isotopic values are far below the line, stream water during the period is dominated by source waters from snowmelt and perhaps from higher elevations as well. If the isotopic values are far above the line, stream water likely experiences strong evaporation or a shift in source waters to lower elevations.






5.2.2. Determining mean elevations of source waters for springs and groundwater

Information on recharge areas of springs and groundwater is paramount for the protection of their quantity and quality (*e.g., Yanggen and Born*, 1990) and for the assessment of climate change effect (*Taylor et al*., 2013), but usually remains unknown in most catchments (e.g., *Chen et al*., 2004) or a challenge (*Koeniger et al*., 2017). Using the isotope-elevation relation (Figure 7), the mean elevations of source waters (recharges) were calculated for springs and groundwater in the mid Merced River catchment (Figure 11), following the same approach as *Jeelani et al.* (2010). For example, the mean source water elevation for Fern Spring was 2,035 m based on its mean $\delta^2H$ values in Table 1 and the equation shown in Figure 7b. This calculation was verified by 30-m DEM using a GIS. The geographic location of Fern Spring was used as a pour point to delineate a drainage area following the same procedure as for groundwater at Hodgdon Meadow and Crane Flat. The mean catchment elevation determined with DEM is 2,108 m for Fern Spring (its catchment ranging in elevation from 1,199 m to 2,277 m). The difference in the mean catchment elevation between the two methods is only 73 m, which is less than $1\sigma$ value determined by the isotope-elevation relation (Figure 11). The mean source water elevation for Drinking Fountain, which was calculated to be 1,014 m by the isotopic approach, can also be verified anecdotally. Drinking Fountain (372 m) is located between Sweetwater Creek and Bear Creek in the low mountain areas (Figure 1). The mean drainage elevation determined by DEM is 1,058 m for Sweetwater Creek and 913 m for Bear Creek, which are slightly higher and lower, respectively, than the mean source water elevation of Drinking Fountain determined by the isotope method. These results demonstrate the reliability of the isotopic method and further validate the isotope-elevation relationship established using small streams, rock glacier outflows, and groundwater, as these sites were not included in the analysis of isotope-elevation relationship.

Based on the $\delta^2H$-elevation relation, the mean source water elevation for springs at Happy Isles and Fen in Yosemite Valley is higher than 2,500 m, approximately 1,500 m above their resurfacing locations (Figure 11). The mean source water elevation is close to 2,500 m for deep wells in Yosemite Valley and to 2,000 m for shallow wells at El Portal. The mean source water elevations for these springs and groundwater are around the present and future threshold elevations (2,181 m for 1995-2004 and 2,486 m for 2085-2094 in Sierra Nevada) determined by *Scalzitti et al*. (2016), below which the variability of snowpack is primarily determined by temperature and above which by precipitation. The source waters of these springs and groundwater will likely be



subject to the impact of both temperature increase and precipitation pattern change in the future. These springs, including Fern Spring, one of the most attractive touring sites in the valley, could be negatively impacted by the shift in snow-rain proportion in the future as their recharge areas

are centered in the upper snow-rain transition zone. So do the groundwater storage and water table dynamics in both Yosemite Valley and El Portal. However, the response is certainly more sensitive in the valley than in El Portal as the source water area of groundwater in the valley extends from ~1,180 m (where wells are located) to > 2,500 m, with more areas located in the snow-covered area than the source water area of groundwater in EL Portal (which extends from < 500 m to >

2,000 m).

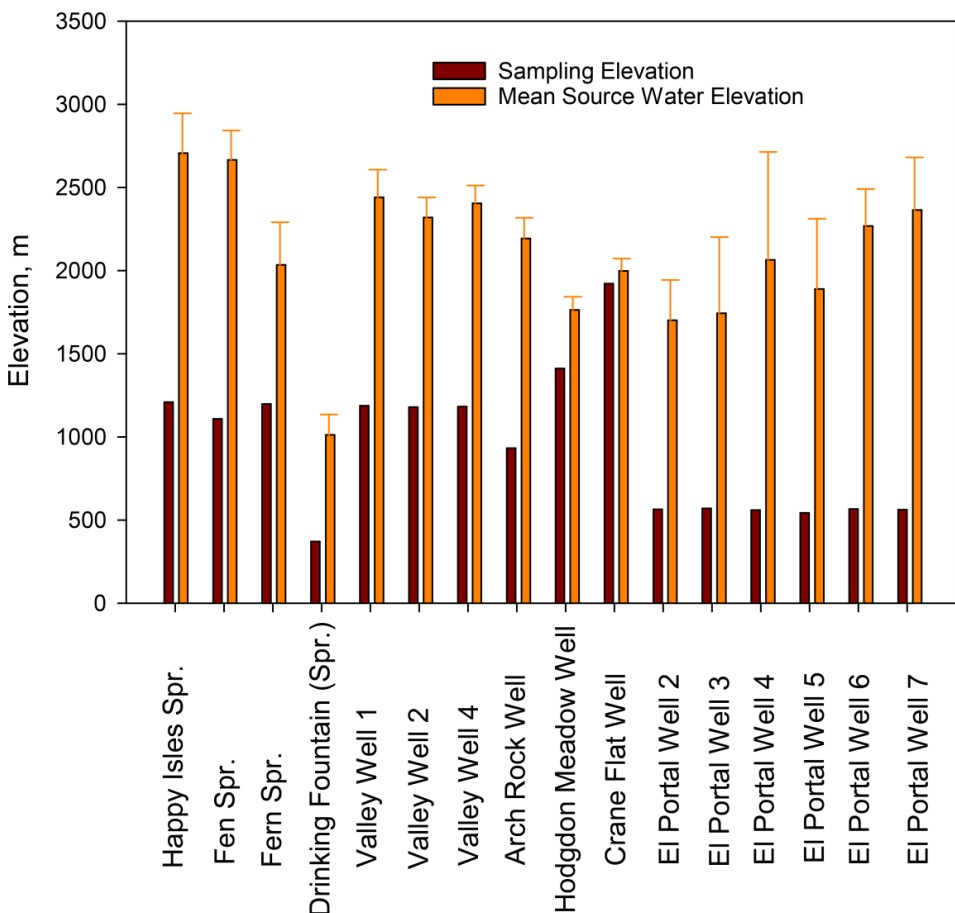

Figure 11. Mean elevations of recharge areas for springs and groundwater calculated by $\delta^2$H-elevation relation, along with 1$\sigma$ standard deviations and sampling elevations.



Note that the estimated source water (recharge) elevations for groundwater in the valley and El Portal refer to elevations where water originated. The path ways of source waters, e.g., whether via direct underground flow paths as of the case in *Frisbee et al.* (2013) or by mixing of groundwater recharge and river water as suggested by *Shaw* et al., (2014), cannot be elucidated by stable isotopic data alone but can be done by combining isotopes and geochemical tracers as

demonstrated by *Liu et al*. (2004). Unlike *Adomako et al*., (2010), in addition, the recharge rates of groundwater and spring water cannot be determined in our study. However, the recharge elevation ranges do help improve our understanding of the sensitivity of climate change impact on groundwater recharge.

**6. Conclusions**

      Stable isotopic composition of stream water and groundwater is strongly controlled by elevations of source waters in the mid Merced River catchment, with an average isotopic lapse rate of -1.9‰/100m for $\delta^2$H and -0.22‰/100m for $\delta^{18}$O in meteoric water. This lapse rate, determined by small streams and groundwater, is more robust than the one established earlier using snow

samples collected in Sierra Nevada. Temporal variability of isotopic compositions in stream water and groundwater was significantly attenuated compared to that in precipitation. Evaporation had little effect on isotopic signature of precipitation, spring water, and groundwater, but affected stream water particularly during low flows in summer and fall. The isotopic composition of stream water was most depleted during the snowmelt periods, as a result of significant contributions of

snowmelt runoff. However, the isotope-elevation relation was not significantly affected by evaporation and snowmelt effects, nor by isotopic fractionation between ice and liquid water in snowmelt. The isotopic composition in stream water in the Merced River consistently becomes more enriched with decreasing sampling elevations (or increasing in drainage area) for all seasons. Using the isotope-elevation relation, a catchment characteristic isotopic value (CCIV) was

established based on the mean drainage elevation. CCIV, in combination with local meteoric water line and evaporation line, helps elucidate the hydrometeorologic processes at different stages or seasons and the sensitivities of stream flow in response to climate warming. The analysis suggests that Yosemite Creek is most sensitive to climate warming due to strong evaporation associated with waterfalls. It is also suggested that evaporation effect on stream flow must be considered in

understanding how climate change impacts stream flow. Based on the isotope-elevation relation,





it was determined that groundwater in the valley is from drainage areas centered in the upper snow-rain transition zone (2,000–2,500 m). It is suggested that groundwater (including spring water) in the valley is very vulnerable to the shift in snow-rain ratio. Continuous and frequent monitoring of changes in stable isotopes in stream water and groundwater along an elevation gradient is a very

powerful tool in watershed hydrology for major snowmelt-fed river systems in the region such as the U.S. West, which will greatly help advance our understanding of how stream flow responds to temperature rise and shift in snow-rain ratio.

**Author contribution**


FL and MC designed the experiments and FL and GS carried them out. FL performed data analyses and developed all figures and tables. FL prepared the manuscript with contributions from all co-authors.

**Competing interests**

The authors declare that they have no conflict of interest.

**Data availability**


The isotopic data used in this study will be available to the public through CUASHI data repository site. Authors are working on the task and a DOI will be provided by the time of acceptance.

**Acknowledgements**

The authors thank Dr. Robert Rice and Dr. Peter Kirchner for taking rock glacier outflow samples, Katy Warner at the Yosemite National Park for taking precipitation samples, and Denise Melendez for processing meteorological data and analyzing samples. Two undergraduate students, Dannique Aalbu from the University of California, Merced and Clifford Tonsberg Jr. from the University of Tennessee, helped sampling in summer 2008, supported by the NSF's Research

Experience for Undergraduates (REU) program. Funding was primarily provided by California Energy Commission through Public Interest Environmental Research (No. 500-02-004) and Dr.





Martha Conklin's start-up fund at the University of California, Merced and Dr. Fengjing Liu's start-up fund from Michigan Technological University. The Research was also supported in part by the National Science Foundation, through the Southern Sierra Critical Zone Observatory (EAR-0725097 and EAR1331939), and the USDA-NIFA Capacity Building Grant Program (#2011-38821-30956 and #2013-38821-21461) and two Evans-Allen Grants (#0225140 and #1007239).

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
