# Peer review of "Elevational Control of Isotopic Composition and Application in Understanding Hydrologic Processes in the mid Merced River Catchment, Sierra Nevada, California, USA"

_Hydrology and Earth System Sciences, 2023_

## Referee Comment (RC1)

**Review on manuscript hess-2023-230**

Title: Altitudinal Control of Isotopic Composition and Application in Understanding Hydrologic Processes in the mid Merced River Catchment, Sierra Nevada, California, USA

Author(s): Fengjing Liu et al.

**General comments:**

In the manuscript by Liu et al. the authors aimed at quantifying how changes in the snow-rain proportion affect stream flow and groundwater recharge in a snowmelt-fed river system. Study site was the mid Merced River catchment, representative for the central and southern Sierra Nevada.
In extensive field campaigns hydrologic and meteorological data of precipitation, snow depths, daily mean discharge were collected, as well as water samples from the main (Merced) River and tributaries, springs, groundwater, snowpits, and glacier melt. Isotope data were available from stream water, groundwater, and springs weekly to biweekly (to monthly) from 2005-2008.
With their comprehensive study they try to better understand the processes or factors that control the spatiotemporal variability of isotopic composition in precipitation, stream water, and groundwater and how such information could be used to advance the understanding of hydrometeorologic and hydrologic processes. The manuscript is very well structured and nicely written. The topic fits well to the scope of the journal and appears to be of interest in the catchment hydrology and alpine hydrology community.

Based on the hydrologic and meteorological data they could show that less snow and earlier snowmelt lead to a shift in peak river runoff toward late winter and early spring, away from summer when water demand is highest. Based on the isotope data, they implemented a catchment characteristic isotopic value (CCIV) in order to elucidate hydrometeorologic processes over seasons – an interesting approach which seems to be quite appropriate for snowmelt-fed catchments.

However, changes or systematic shifts in the snow-rain transition zone due to climate change couldn't really be proved since the observation period was too short and further, during their 3-year observation period they stated that one of the years was very wet and one very dry. Therefore, only relying upon the data from these extreme years seems to be questionable or it might at least be difficult to draw reliable conclusions, especially long-term conclusions on climate change. It would thus be great to continue the time series in the future.

**Specific Comments:**

P7, Table 1: D-Ex data are missing – perhaps due to temperature issues for d18-O during isotope analysis with the DLT-100?

P9, l. 196 please insert standard deviation after 1σ - in order to define this acronym in the first instance

P9, l. 246 DEM - define this acronym in the first instance

P9, l. 255 WY - define this acronym in the first instance

P16, l. 389 LMWL - be careful when establishing a LMWL based on snow samples because of isotopic fractionation.

P28, Figure 10b and text p. 29 l. 683ff: I'm not sure that evaporation in Yosemite is much stronger… True, it plotted further right but the slope seems to be steeper than for Tenaya.

---

## Author Response (AR1)

**Responses to Reviewers' Comments**

Thanks to the editor and two anonymous reviewers for their time spent reviewing our manuscript and general support for our study! We revised our manuscript following their comments and suggestions, where appropriate. In addition, all figures, tables, their captions, and main text were checked and made sure error free. Below are our point-to-point responses (**AR** for author response) to the comments and suggestions provided by two reviewers. I hope our revision is satisfactory to all.

**Anonymous Referee #1, 07 Dec 2023**

General comments:

In the manuscript by Liu et al. the authors aimed at quantifying how changes in the snow-rain proportion affect stream flow and groundwater recharge in a snowmelt-fed river system. Study site was the mid Merced River catchment, representative for the central and southern Sierra Nevada.

In extensive field campaigns hydrologic and meteorological data of precipitation, snow depths, daily mean discharge were collected, as well as water samples from the main (Merced) River and tributaries, springs, groundwater, snowpits, and glacier melt. Isotope data were available from stream water, groundwater, and springs weekly to biweekly (to monthly) from 2005-2008.

With their comprehensive study they try to better understand the processes or factors that control the spatiotemporal variability of isotopic composition in precipitation, stream water, and groundwater and how such information could be used to advance the understanding of hydrometeorologic and hydrologic processes. The manuscript is very well structured and nicely written. The topic fits well to the scope of the journal and appears to be of interest in the catchment hydrology and alpine hydrology community.

Based on the hydrologic and meteorological data they could show that less snow and earlier snowmelt lead to a shift in peak river runoff toward late winter and early spring, away from summer when water demand is highest. Based on the isotope data, they implemented a catchment characteristic isotopic value (CCIV) in order to elucidate hydrometeorologic processes over seasons – an interesting approach which seems to be quite appropriate for snowmelt-fed catchments.

**AR:** Thanks for the summaries and supporting comments!

However, changes or systematic shifts in the snow-rain transition zone due to climate change couldn't really be proved since the observation period was too short and further, during their 3-year observation period they stated that one of the years was very wet and one very dry. Therefore, only relying upon the data from these extreme years seems to be questionable or it might at least be difficult to draw reliable conclusions, especially long-term conclusions on climate change. It would thus be great to continue the time series in the future.

**AR**: The future shift of more rainfall relative to snowfall was not part of this study, but the trend in the US West has been well documented by many other studies such as those cited in our manuscript (e.g., Mote et al., 2005; Knowles et al., 2006; Stewart et al., 2004). Under this backdrop, two objectives were specified, as described in the text: One, to understand the factors that control spatial variation of isotopic values in stream flow and groundwater and the other, which depends on the first, to demonstrate the applications to improve our understanding in hydrology and hydrometeorology and implications to infer climate change impacts on stream flow and groundwater recharge using space-for-time concept. Among the three years studied, two of them were extreme and the third was mild, which provides an excellent opportunity for us to test, for example, if the isotopic lapse rates vary over seasons with very different climates. We agree that it would be invaluable to continue the monitoring and extend the time series of the data. With an extended time-series of the data, we may be able to do much more, e.g., to examine trends in real time, as this reviewer envisioned, and quantify the actual rate of changes in groundwater recharge and evaporation effects.

Specific Comments:

P7, Table 1: D-Ex data are missing – perhaps due to temperature issues for δ18-O during isotope analysis with the DLT-100?

**AR**: That is our concern, as well. For this reason, we consistently relied on the slope of 2H-18O, which utilized the information from both 18O and 2H together more qualitatively than quantitatively to minimize the impact of 18O analysis.

P9, l. 196 please insert standard deviation after 1σ - in order to define this acronym in the first instance

**AR**: Accepted as suggested (see P9, L236-237 in the revised version)

P9, l. 246 DEM - define this acronym in the first instance

**AR**: Accepted as suggested (see P9, L244-245 in the revised version).

P9, l. 255 WY - define this acronym in the first instance

**AR**: Defined as suggested in the text (see P9, L256 in the revised version).

P16, l. 389 LMWL - be careful when establishing a LMWL based on snow samples because of isotopic fractionation.

**AR**: Agreed, but our samples were collected at the peak of snow accumulation. Still, we do not mean it is free of isotopic fractionation, but that effect should be minimized and the isotopic signature in snow at the maximum accumulation mostly represents the signature of input water to the hydrologic system.

P28, Figure 10b and text p. 29 l. 683ff: I'm not sure that evaporation in Yosemite is much stronger… True, it plotted further right but the slope seems to be steeper than for Tenaya.

**AR**: A good catch, thank you! The figure panel (10b) was rescaled on both x- and y-ordinates before the submission and accidentally hid the most important data points, which are the samples collected in Yosemite Creek near the end of flow seasons from 2006 to 2008. They are scaled back now and highlighted with an oval circle and date ranges, along with the slope and intercept of Yosemite Creek samples collected near the end of flow seasons. A significant 18O-2H relationship cannot be established for Teneya Creek samples collected during the same periods, likely due to small variation in 18O and 2H relative to their analytical accuracies.

**Anonymous Referee #2, 10 Dec 2023**

The study conducted intensive sampling for isotopic compositions in precipitation, stream water and groundwater to help understand the hydrological processes in a typical snowmelt-fed catchment. The topic is very interesting and valuable and the manuscript is well organized.

**AR:** Thanks for the support!

However, my major concern for this study is that the objective is very vague. I couldn't catch what are the key points of the study. The elevational control of isotopic composition is a basic law for isotope hydrology and the phenomenon has been widely reported. The effect of evaporation effect makes stable isotope of streamflow changes among tributaries is not unusual. Thus, it should further focus on more specific issues and new findings or contributes to hydrology from the detailed samplings instead of presenting the form of the research as a form of case study. I suggest put more attention on the identification the recharge zones of groundwater and streamflow and their changes or the impact of snowpack decline on the recharge of groundwater and streamflow.

**AR**: In paragraph 3, we started with the point that stable isotopes are helpful to resolve the issues introduced in the first two paragraphs. Then, we talked about the conventional, broad uses of stable isotopes to highlight three points: (1) one attribute may be useful for one use but may pose a challenge to the other; (2) it is thus important to understand the factors/processes that control the spatiotemporal variation of isotopic composition that are relevant to the studied subject (thus justifying our first objective); and (3) what else we can solve using stable isotopes in watershed hydrology. However, it is difficult to justify (3) in details or by publications, as we do not know what else we can use for until we show and examine our data. Other than determining the recharge elevations using the isotopic lapse rate, other uses are novel and thus we cannot define them in detail *a priori*. Instead, we framed them in a more general term (what we can do to improve our understanding in hydrology and hydrometeorology). Following the suggestion here, we revised our introduction and added more details on the isotopic lapse rate and recharge elevations and brought some specific research questions into the context, as the quality of isotope-elevation relationship is not always guaranteed. For example, how can we determine the isotope-elevation relationship (e.g., using precipitation samples vs stream samples as we discussed in the study)? Do evaporation and isotopic fractionation affect the relationship? How

does the relationship vary over seasons and years, particularly with very different climates? These are important questions we have to examine before we apply the relationship in our, and perhaps any, studies.

Specific comments:

Lines 5-10. The title is too long. "Altitudinal" should be "elevational".

**AR**: The title is indeed a bit lengthy (21 words), but we really want it to reflect our two major objectives in the title. We changed "altitudinal" to "elevational", as suggested, to make it consistent with "elevation" and "elevational" used in the main text.

Lines 41-43 "flow and flow duration of Yosemite Creek are much more sensitive to temperature increase due to a strong evaporation effect caused by waterfalls, suggesting possible prolonged dry-up period of Yosemite Falls in the future." The conclusion seems arbitrary.

**AR**: We think the comment likely arises from concern about Figure 10(b). With the correction of the error on Figure 10(b), we are confident on the result. However, we still changed "are" to "appear to be" in response to the reviewer's concern.

The Section of Introduction seems too general and the part should be re-organized to review the state-of-the-art methods on identifying the streamflow and groundwater sources and recharge zones, or how the stable isotopes could be used to improve our understanding on the certain hydrological process in complex catchment.

**AR**: Modified as described above in response to the general comment.

Section 2. What's the annual runoff depth of the catchment and is there significant variation of precipitation and runoff depth with elevation? What is the temperature lapse rate? Besides, I suggest presenting the time series of precipitation. It may be added in Fig 2.

**AR**: We added available data following the suggestions, including temperature lapse rate and the change of SWE over elevation for 2004-2005 from others' study in the section (please see P5, L141-145 in the revised version). The change of annual precipitation with elevation was implicitly shown in Figure 2a, where annual precipitation is shown as the last point of annual cumulative precipitation and elevation is marked following each station's name, but a gradient cannot be really calculated because there are just three stations with complete data sets for only two years and they all are located in lower elevations. Providing time series for precipitation makes it really messy and does not clearly indicate the variation among years and general pattern in change of precipitation with elevation. The general trend of temporal variation of precipitation can be seen from annual cumulative figure. In addition, time series of precipitation is not really crucial for the study.

Figure 2. The annual cumulative precipitation is large at Gin Flat, but the snow depth is small. Is it a mistake?

**AR**: Other than Gin Flat, the other two sites shown on 2(a) and 2(b) are different sites, with different elevations. SnoTel sites do not usually provide annual precipitation data, while meteorological stations may not have snow depth measurements. We presented whatever was available for the catchment. We added an annotation in the caption of Figure 2 to indicate the difference in sites between 2(a) and 2(b).

Figure 4. I suggest using a different line type to show PSF to make the figure more clearly.

**AR**: A dashed line is used for PSF now to highlight the peak flow.

Lines 464-468. It's not clear how the evaporation effect correction is conducted.

**AR**: As described in the text (please see P20, L463-466 in the revised version) and the caption of Table 2, the intersection (point) between LMWL and LEL was used as the correction. Mathematically, it is the solution of two simultaneous equations of LMWL and LEL. We annotated the mathematical approach under Table 2.

In Figs 5 and 8, the significance test is needed.

**AR**: Changed as suggested.

Lines 682-685, the slope of lines for Tenaya Creek seems lower than Yosemite Creek in Fig 10. It's hard to say the evaporative effect is more intensive in Yosemite Creek.

**AR**: A good catch, thank you! The figure panel (10b) was rescaled on both x- and y-ordinates before the submission and accidentally hid the most important data points, which are the samples collected in Yosemite Creek near the end of flow seasons from 2006 to 2008. They are scaled back now and highlighted with an oval circle and date ranges, along with the slope and intercept of Yosemite Creek samples collected near the end of flow seasons. A significant 18O-2H relationship cannot be established for Teneya Creek samples collected during the same periods, likely due to small variation in 18O and 2H relative to their analytical accuracies.

---

## Author Response (AR2)

**Responses to Reviewers' Comments/Suggestions**

Thanks to the editor and two anonymous reviewers for your time spent on reviewing our manuscript and support for our study! We revised our manuscript following the comments and suggestions provided by reviewers, where appropriate. Below are our point-to-point responses (**AR** for author response) to the comments and suggestions from reviewers and journal staff who make the quality control. I hope our revision this time is satisfactory to all.

**(1) Notification to the authors by Polina Shvedko:**

The title page of *pdf. manuscript file must include the full institutional addresses of all authors. However, country name is missing from the affiliations. Please add it for the next revision.

**AR**: Added as suggested. We also added our zip codes.

**(2) Report #1**

Accepted as is

**AR**: Thanks Reviewer 1 for your continued support of our study!

**(3) Report #2**

The paper focuses on the elevational control of isotopic composition and application in understanding hydrologic processes. The topic is very interesting and valuable and the manuscript is well organized.

**AR**: Thanks Reviewer 2 for your continued support of our study!

However, there are still several problems that need to be addressed.

**AR**: We revised our manuscript again following the comments and suggestions provided and we justified where we think a change is not necessary.

Lines 38-39. "catchment-characteristic isotopic value" should explain simply.

**AR**: Added a phrase to explain the term (see P1, L38-39 in the revised version).

Lines 42-43. "more sensitive to temperature increase" indicates more sensitive to seasonal temperature increase during the baseflow period. It should be revised to remove misunderstanding with temperature increase with climate change

**AR**: Changed as suggested (P1, L43-44 in the revised version).

Line 50. It is not suitable to use climate change as a key word. Further, climate change should not be emphasized in the manuscript and the use of the word in the main body should be cautious as the topic of the manuscript is indirectly related to the climate change.

**AR**: Removed as a key word.

Lines 283-290. Annual runoff depth for the gauged catchments is suggested to show the general hydrologic conditions.

**AR**: The reviewer got a good point. For our purpose, however, flow rate was consistently used in both figures and text in the manuscript. Using flow rate, the rainfall effect on flow in the lower elevation gage (Briceburg) can be clearly seen and easily compared with the high-elevation gages. In particular, the measurement errors at Briceburg can be clearly spotted. Using runoff depth, however, the curves were intertwined, and the above effects were not easily spotted.

Lines 627-630. CCIV is a core value of the manuscript. Details on the calculation are needed. The application of CCIV is an important part for the manuscript. It may be better to put it to the result part and put more attention on the application of CCIV for hydrological processes, like determining the duration and the magnitude of snowmelt events.

**AR**: We are delighted that Reviewer 2 really likes the CCIV section. Equally, a group of mountain hydrologists were very excited about the modeling of groundwater recharge elevations when we presented our work in a hydroclimate conference in California, USA. To put the CCIV section in the Results, it is assumed that we knew such a characteristic/phenomenon exists *a priori*. As a matter of fact, we were able to demonstrate its usefulness only after we extensively discussed the controlling factors on isotopic composition in streamflow in the first part of the Discussion section. Thus, we argue that it is the best to have it in the Discussion section, together with the modeling of groundwater recharge elevations. Also, we added a sentence to explain how to calculate CCIV (P26, L630-631 in the revised version).

Lines 699-700. "the arithmetic mean isotopic value from samples" may be better. More explanation is needed for lines 703-705.

**AR**: Two great suggestions here from the reviewer! Arithmetic mean is indeed what we meant (thus added as suggested). The argument about Yosemite Creek was enhanced by adding specific details (see P29, L704-705 in the revised version).

Lines 761-770. It should also be noted that the method assumes that the difference of runoff depth generated at different elevation is ignored.

**AR**: It was added as a clause in the argument (see P32, L768-769 in the revised version).